# A KLF6-driven transcriptional network links lipid homeostasis and tumour growth in renal carcinoma

Saiful E. Syafruddin[1,2], Paulo Rodrigues[1], Erika Vojtasova[1], Saroor A. Patel[1], M. Nazhif Zaini[1], Johanna Burge[3], Anne Y. Warren[4], Grant D. Stewart[3], Tim Eisen[5,6,7], Dóra Bihary[1], Shamith A. Samarajiwa [1] & Sakari Vanharanta[1]

Transcriptional networks are critical for the establishment of tissue-specific cellular states in health and disease, including cancer. Yet, the transcriptional circuits that control carcinogenesis remain poorly understood. Here we report that Kruppel like factor 6 (KLF6), a transcription factor of the zinc finger family, regulates lipid homeostasis in clear cell renal cell carcinoma (ccRCC). We show that KLF6 supports the expression of lipid metabolism genes and promotes the expression of *PDGFB*, which activates mTOR signalling and the downstream lipid metabolism regulators SREBF1 and SREBF2. *KLF6* expression is driven by a robust super enhancer that integrates signals from multiple pathways, including the ccRCC-initiating VHL-HIF2A pathway. These results suggest an underlying mechanism for high mTOR activity in ccRCC cells. More generally, the link between super enhancer-driven transcriptional networks and essential metabolic pathways may provide clues to the mechanisms that maintain the stability of cell identity-defining transcriptional programmes in cancer.

[1] MRC Cancer Unit, University of Cambridge, Hutchison/MRC Research Centre, Box 197, Cambridge Biomedical Campus, Cambridge CB2 0XZ, UK. [2] UKM Medical Molecular Biology Institute, Universiti Kebangsaan Malaysia, Jalan Yaacob Latiff, Bandar Tun Razak, Kuala Lumpur 56000, Malaysia. [3] Academic Urology Group, Department of Surgery, University of Cambridge, Addenbrooke's Hospital, Cambridge Biomedical Campus, Cambridge CB2 0QQ, UK. [4] Department of Histopathology, Cambridge University Hospitals NHS Foundation Trust, Cambridge CB2 0QQ, UK. [5] Department of Oncology, University of Cambridge, Cambridge CB2 0XZ, UK. [6] Department of Oncology, Addenbrooke's Hospital, Cambridge University Health Partners, Cambridge CB2 0QQ, UK. [7] Oncology Early Clinical Development, AstraZeneca, Cambridge SG8 6EH, UK. Correspondence and requests for materials should be addressed to S.V. (email: sv358@mrc-cu.cam.ac.uk)

Renal cancer is responsible for >400,000 new diagnoses and 140,000 deaths annually worldwide[1]. The most common form of renal cancer, clear cell renal cell carcinoma (ccRCC), accounts for ~75% of all renal cancers[2]. Biallelic inactivation of the *von Hippel-Lindau (VHL)* tumour suppressor gene is a hallmark event in ccRCC pathogenesis, contributing to ~90% of sporadic cases[3] as well as to hereditary ccRCC in von-Hippel-Lindau syndrome patients[4]. The VHL protein mediates proteasomal degradation of the hypoxia-inducible factor (HIF) alpha subunits under normoxic conditions, and genetic *VHL* inactivation in ccRCC leads to constitutive HIF alpha accumulation and consequent upregulation of hypoxia-associated genes[4]. Of the two major HIF alpha subunits, HIF2A is responsible for driving ccRCC growth while HIF1A may suppress ccRCC progression[4,5]. Histologically, ccRCCs are hyper-vascular due to upregulation of pro-angiogenic factors such as *VEGFA*, a downstream target of the hypoxia pathway. Hence, the development of ccRCC therapies has focused on inhibiting angiogenesis, and several anti-angiogenic drugs targeting receptor tyrosine kinases (RTKs) such as VEGFR and PDGFR have been clinically approved for ccRCC[6]. Additionally, small molecule inhibitors targeting HIF2A have been developed with promising results in some ccRCC patients[7–9].

The mTOR pathway, frequently hyper-activated in ccRCC[10,11], is another clinically relevant target in renal cancer. Two approved mTOR inhibitors, everolimus and temsirolimus, are licensed for clinical use in ccRCC patients[12,13]. However, the mechanisms underlying increased mTOR activity in ccRCC remain incompletely understood. Genetic analysis of ccRCC has identified mutations in components of the PI3K-AKT-mTOR signalling cascade[14]. For example, *PIK3CA* and *PTEN* are mutated in 2–5% of ccRCCs and some mutations have also been observed in *TSC1*, a negative regulator of mTOR[15]. Furthermore, mutations in *MTOR* are found in approximately 6% of ccRCCs[14,16]. Genetic alterations are thus likely to contribute to mTOR activation in ccRCC, although upstream activating signals still seem to be required in most cases[16]. The recent generation of double knockout *Vhl*−/−*;Pbrm1*−/− and *Vhl*−/−*;Bap1*−/− mouse models have also identified mTORC1 hyper-activation as a potential driver of ccRCC[17,18]. Concomitant loss of *VHL* and either *PBRM1* or *BAP1*, two frequently mutated ccRCC genes[3,14] could thus directly lead to increased mTORC1 activity during ccRCC development. Mutational analyses alone have not, however, been sufficient to identify patients sensitive to mTOR inhibition[19], suggesting that additional molecular players are involved.

Recent multi-region sequencing efforts have described widespread intratumoral genetic heterogeneity in ccRCC[3]. Apart from large chromosome 3p deletions and inactivation of the second allele of *VHL*, usually by point mutation, most predicted ccRCC driver mutations are subclonal[3]. This suggests that the opportunities for mutation-based therapeutic approaches in ccRCC are limited to dependencies resulting from inactivation of the VHL pathway. Given that a significant fraction of ccRCCs display intrinsic resistance to HIF2A inhibition and anti-angiogenic agents[9,20], a better understanding of the molecular dependencies of *VHL* mutant ccRCC is needed. To this end, tissue-specific transcriptional circuits or lineage dependencies could offer a viable avenue forward[21].

The expression of transcriptional regulators that govern key biological processes such as cell identity and cell fate is often associated with large enhancer clusters such as super enhancers[22,23]. Super enhancers also regulate cancer phenotypes[24,25]. In this study, combining chromatin activation and transcriptomic data from multiple ccRCC model systems and clinical samples, we find that one of the strongest super enhancers in ccRCC cells, partially activated by the ccRCC-initiating VHL-HIF2A pathway, is

associated with the *KLF6* locus, a gene encoding a zinc finger DNA-binding transcription factor of the Kruppel-like family. KLF6 inhibition impairs ccRCC fitness and leads to a profound inhibition of lipid biosynthetic pathways. KLF6 regulates the expression of several lipid homeostasis genes. Moreover, by supporting the expression of *PDGFB*, an agonist of the mTOR pathway, KLF6 further promotes lipid metabolism by enhancing the activation of the key lipid metabolic regulators SREBF1 and SREBF2. KLF6 and mTORC1 thus co-regulate lipid homeostasis, consequently supporting ccRCC growth. The PDGF and mTOR pathways are clinically relevant therapeutic targets in ccRCC. Our data provide a molecular link between these pathways, and gives clues to their mechanism of action as drivers of ccRCC. Furthermore, the functional connection between super enhancer-driven transcriptional programmes and core metabolic pathways may help explain the stability of cell identity-defining molecular networks in cancer.

## Results

**KLF6 is associated with a super enhancer in ccRCC.** Histone H3 lysine 27 acetylation (H3K27ac) marks active gene regulatory elements[26], which in some genomic loci cluster together to form large domains or super enhancers characterized by particularly high H3K27ac signal[23]. Several lines of evidence suggest that super enhancers promote the expression of critical transcriptional regulators in various biological contexts[22,23]. In order to identify transcriptional networks that support ccRCC progression, we analysed H3K27ac chromatin immunoprecipitation coupled with high-throughput sequencing (ChIP-seq) data from several *VHL* mutant ccRCC cell lines[27] and looked for transcription factor-associated super enhancers. We found that one of the strongest super enhancers in ccRCC cells encompassed *KLF6*, a gene encoding a zinc finger DNA-binding transcription factor (Fig. 1a). Similarly, high H3K27ac signal was also observed in the proximity of the *KLF6* locus in ccRCC patient samples and ccRCC xenografts (Fig. 1b). In line with the possibility that the super enhancer regulates *KLF6*, we observed high and relatively specific expression of *KLF6* in ccRCC samples when compared to other solid cancer types in the large TCGA cohort (Supplementary Fig. 1a). *KLF6* expression was also higher in ccRCC samples when compared to normal kidney tissue (Supplementary Fig. 1b), and ccRCC cell lines, including highly metastatic derivatives[28], expressed high levels of KLF6 protein (Supplementary Fig. 1c).

**KLF6 supports ccRCC fitness in vitro and in vivo.** *KLF6* can be expressed as several differentially spliced variants (SV-1, SV-2 and SV-3), some of which have been linked to tumour progression[29,30]. We analysed RNA-seq data from several ccRCC cell lines to determine the expression level of the full-length *KLF6* as well as the reported three *KLF6* variants. Full-length *KLF6* was the predominant isoform and we found little evidence for the expression of the other variants (Supplementary Fig. 1d). To test the biological relevance of KLF6, we inactivated KLF6 using lentivirally delivered CRISPR-Cas9 in *VHL* mutant ccRCC cell lines. We used cell lines derived from human tumours (UOK101 and RCC-MF) but also experimentally derived highly metastatic subclones of human ccRCC cell lines (786-M1A and OS-LM1)[28], which recapitulate several important features of human ccRCC at both phenotypic and molecular levels, including high metastatic potential and relevant histology in xenograft assays[28], in vivo response to drug treatment[7], mutational landscape[31], and activation of genes that correlate with poor patient outcome in clinical ccRCC data sets[27,28]. Two *KLF6*-targeting sgRNAs, sgKLF6-4 and sgKLF6-5, were highly efficient in inducing mutations at their predicted target regions (Supplementary Fig. 2a), with corresponding KLF6 protein depletion in the

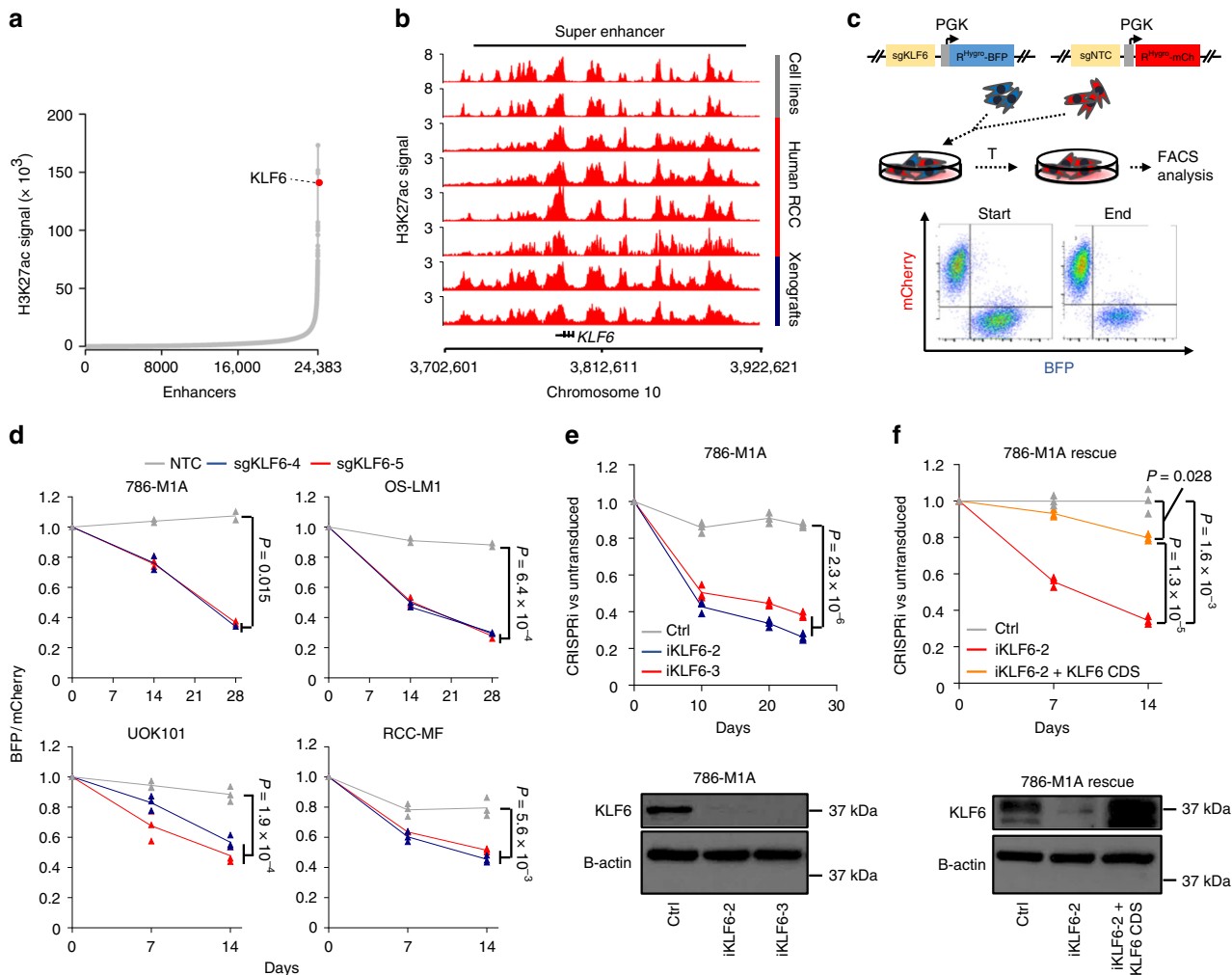

**Fig. 1** KLF6, a super enhancer-associated transcription factor, supports ccRCC growth in vitro. **a** A strong super enhancer, active in ccRCC cells, is proximal to the *KLF6* locus. **b** H3K27ac ChIP-seq signal at the large enhancer cluster in the proximity of the *KLF6* locus in ccRCC cell lines, tumour xenografts and clinical ccRCC samples. **c** Strategy for the competitive proliferation assay. **d** Competitive proliferation assay of KLF6-targeted VHL mutant ccRCC cells (pools of lentivirally transduced CRISPR-Cas9 knock-out cells). The relative fraction of BFP$^+$ KLF6-targeted and mCherry$^+$ control cells, normalized to day 0. 786-M1A and OS-LM1 average of two technical replicates; UOK101 and RCC-MF average of three technical replicates. Two-tailed Student's *t*-test. **e** (Top) Competitive proliferation assay of the population of KLF6-targeted 786-M1A cells (lentivirally delivered CRISPRi). The relative fraction of iKLF6 and NTC transduced cells, normalized to day 0, compared to untransduced cells. Average of three technical replicates. Two-tailed Student's *t*-test. (Bottom) Immunoblot showing KLF6 expression in the cells used in the competitive proliferation assay. **f** (Top) Competitive proliferation assay of KLF6-targeted 786-M1A cell pools (lentivirally delivered CRISPRi) transduced with exogenous KLF6. Relative fraction of iKLF6, iKLF6 with exogenous KLF6 or NTC transduced cells, normalized to day 0, compared to untransduced cells. Average of three technical replicates. Two-tailed Student's *t*-test. (Bottom) Immunoblot showing KLF6 expression in the cells used in the competitive proliferation assay

CRISPR-Cas9-targeted cell pools (Supplementary Fig. 2b). The proliferative capacity of the KLF6 depleted cell pools was then assessed using a competitive proliferation assay (Fig. 1c). In all four cell lines tested, KLF6 depletion led to impaired growth (Fig. 1d). To confirm that the result was not confounded by the fluorescent markers, the assay was also performed by swapping the vector backbones. We observed consistent impaired growth in the KLF6 depleted cells (Supplementary Fig. 2c). Reintroduction of exogenous KLF6 mitigated the proliferative defect (Supplementary Fig. 2d).

In some contexts, CRISPR-Cas9-targeted mutagenesis reduces cell fitness independently of the target locus[32]. We, therefore, validated our findings using lentivirally delivered CRISPR interference (CRISPRi)[33], a non-mutational method of *KLF6* silencing that utilizes a catalytically inactive dCas9 fused with the KRAB transcriptional repressor domain (Supplementary Fig. 2e).

Stable CRISPRi-mediated inhibition of *KLF6* expression in a pool of 786-M1A cells also resulted in impaired ccRCC cell proliferation in vitro, and reintroduction of exogenous KLF6 rescued the phenotype (Fig. 1e, f). We then investigated the phenotype of *KLF6* inhibition in vivo. We injected pools of 786-M1A and OS-LM1 cells in which *KLF6* had been targeted using CRISPR-Cas9 subcutaneously into immunocompromised mice and followed tumour formation and growth over time. *KLF6*-targeted cells formed smaller tumours when compared to control cells (Fig. 2a, b and Supplementary Fig. 3a–c). Genetic analysis revealed selection for the wildtype *KLF6* gene in KLF6-targeted tumours (Supplementary Fig. 3d) when compared to the same cells in tissue culture (Supplementary Fig. 2a), indicating that the tumours formed by KLF6-targeted cells consisted at least partially of escaper cells. Similarly, an experimental lung colonization assay demonstrated reduced metastasis re-initiation capacity in

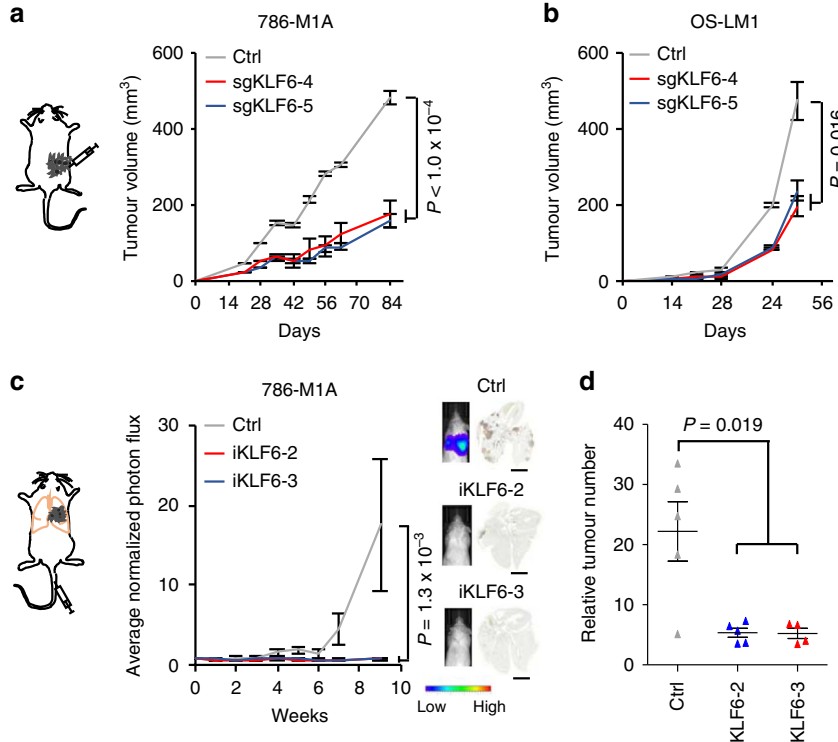

**Fig. 2** KLF6 supports ccRCC growth and metastatic colonization in vivo. **a, b** Subcutaneous tumour growth of KLF6-targeted 786-M1A (**a**) and OS-LM1 (**b**) cells. Pools of lentivirally transduced CRISPR-Cas9 knock-out cells. 786-M1A, $N = 10$ tumours/group. OS-LM1, $N = 14$ tumours for sgKLF6-4 and sgKLF6-5 groups, $N = 12$ tumours for the control group. Error bars, SEM. Two-tailed Mann–Whitney $U$-test. **c** (Left) Average normalized lung photon flux of intravenously inoculated KLF6-targeted or control 786-M1A cells (lentivirally delivered CRISPRi). $N = 5$ mice/group. Error bars, SEM. Two-tailed Mann–Whitney $U$-test. (Right) Representative bioluminescence images and histological lung sections stained with human vimentin. Scale bar, 5 mm. **d** Quantification of the human vimentin-stained lung metastatic foci from the experiment in **c**. Two-tailed Mann–Whitney $U$-test

786-M1A cells in which *KLF6* was silenced using CRISPRi (Fig. 2c, d). These results demonstrate that KLF6 supports ccRCC fitness in vitro and in vivo.

**KLF6 activation by a robust super enhancer**. Cancer-associated super enhancers can be sensitive to perturbation[24,34]. To test whether the *KLF6* super enhancer locus was also sensitive to perturbations in the activity of its constituent enhancers, we employed a CRISPRi-based approach to inactivate several distinct enhancers within the *KLF6* super enhancer locus using five sgRNA pairs in independent tandem constructs (iSE-1–iSE-5), a method we have previously used for highly efficient and specific targeting of distal enhancers[27]. Using p300 ChIP-seq data as a guide[27], we targeted the sgRNA constructs to the p300 peaks in the valleys of H3K27ac signal, regions considered to be the target sites for transcription factor binding. H3K27ac ChIP-seq confirmed efficient and specific targeting of individual regions within this large cluster of enhancers (Fig. 3a). We did not observe clear interdependencies between the different enhancers. We then assessed *KLF6* expression in the enhancer-targeted cells and found that *KLF6* levels were only subtly affected by the inhibition of some of these enhancers (Supplementary Fig. 4a). Combinatorial targeting of enhancer pairs showed additive effects, but even then the expression of *KLF6* remained fairly strong (Supplementary Fig. 4b), whereas simultaneous targeting of all five enhancers reduced *KLF6* mRNA expression by ~60% (Fig. 3b). We then used CRISPR-Cas9 to genetically remove a 113 kb segment of the super enhancer. In cell populations this resulted in relatively modest downregulation of *KLF6* (Supplementary Fig. 4c), possibly reflecting heterogeneity in targeting efficiency.

We, therefore, isolated single cell-derived clones from this population and performed PCR-based screening for the deletion (Supplementary Fig. 4d). We identified a clone with a putative homozygous deletion, a heterozygous deletion and no deletion for further analyses. In line with the PCR data, H3K27ac ChIP-seq confirmed the reduction in signal in the heterozygous clone, while no signal from the targeted region was detected in the clone carrying the homozygous deletion (Fig. 3c). *KLF6* expression was also progressively reduced in the clones carrying super enhancer deletions (Fig. 3d), with homozygous deletion resulting in a similar effect to that seen in the cells in which five constituent enhancer regions were simultaneously targeted by CRISPRi (Fig. 3b). RNA-seq-based analysis comparing the clone with a homozygous deletion to the clone with no deletion revealed that *KLF6* was the most significantly downregulated gene within the 5 Mb genomic region flanking the 113 kb deletion, supporting the possibility that *KLF6* was one of the main targets of this super enhancer (Fig. 3e). These observations suggest that *KLF6* is regulated by a robust super enhancer, which tolerates CRISPRi-mediated perturbation of the individual constituent enhancers. However, simultaneous inactivation of several constituent enhancers, as well as a large deletion of the super enhancer, resulted in significant *KLF6* downregulation, supporting the idea that this large enhancer cluster functions in a modular manner to drive *KLF6* expression in ccRCC.

**KLF6 interacts with the HIF2A pathway in ccRCC**. Given the prominent role of HIF2A in ccRCC pathogenesis, we next tested whether the VHL-HIF2A axis modulated *KLF6* expression. First, analysis of the ccRCC TCGA data revealed a positive correlation

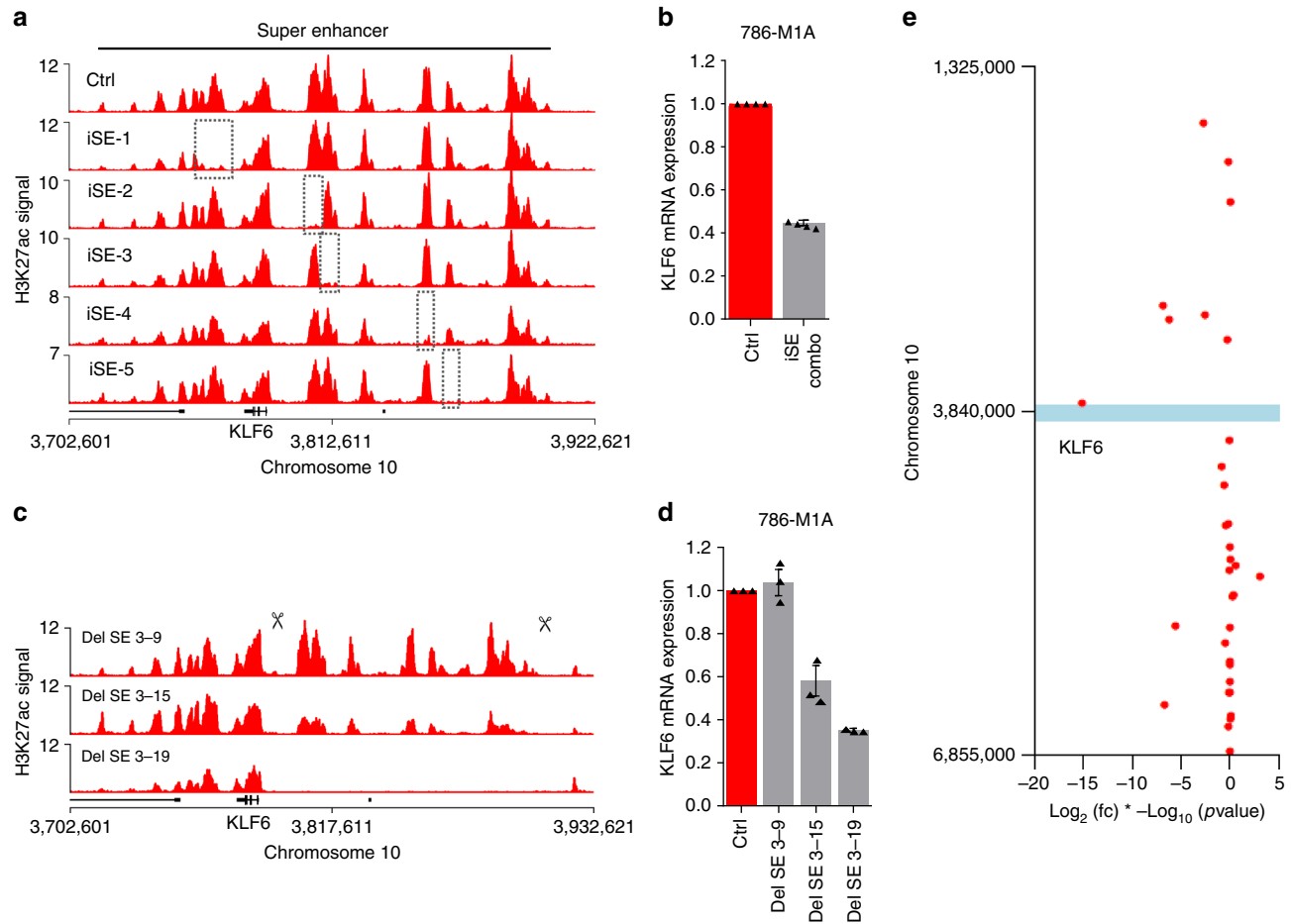

**Fig. 3** KLF6 is regulated by a robust super enhancer. **a** H3K27ac ChIP-seq signal in the *KLF6* locus. The target region for CRISPRi-based enhancer inactivation indicated by dashed boxes. **b** KLF6 mRNA expression in the pool of 786-M1A cells with simultaneous CRISPRi-mediated inactivation of the five enhancer regions shown in **a**. Average of four experiments. Error bars, SEM. **c** H3K27ac ChIP-seq signal in the *KLF6* locus in clones with no deletion (top), heterozygous deletion (middle), and homozygous deletion (bottom). **d** *KLF6* expression in the single cell-derived clones shown in **c**. Average of three experiments. Error bars, SEM. **e** Expression of genes located within a 5 Mb window flanking the 113 kb deletion in the super enhancer region. Clone Del SE 3-19 compared to clone Del SE 3-9 by RNA-seq, $N = 4$ for both samples. The blue bar indicates the location of the 113 kb deletion

between *HIF2A* and *KLF6* expression, similar to that seen for the well-characterized HIF2A target gene *CCND1* (Fig. 4a). Alongside *CCND1* and *CXCR4*, *KLF6* mRNA expression was reduced in VHL-reintroduced 786-M1A and OS-LM1 cells (Fig. 4b). KLF6 down-regulation was also observed at the protein level in both cell lines (Fig. 4c). HIF2A has been recently linked to the maintenance of ccRCC super enhancers[35]. We, therefore, used ChIP-seq to test the effects of VHL reintroduction, and the consequent loss of HIF2A, to the H3K27ac patterns of the *KLF6* super enhancer locus. We observed a reduction in H3K27ac signal at one of the enhancer regions downstream of the *KLF6* locus in 786-M1A and OS-LM1 cells (Fig. 4d). Analysis of HIF2A ChIP-seq data confirmed that HIF2A bound the same enhancer region (Fig. 4e and Supplementary Fig. 5a). We also observed reduced HIF2A binding at this locus upon VHL reintroduction by ChIP-qPCR in 786-M1A and OS-LM1 cells (Supplementary Fig. 5b). However, consistent with the previous observation that this particular super enhancer is insensitive to perturbation (Fig. 3a), the general H3K27ac pattern remained mostly unchanged (Fig. 4d). Using the CRISPRi approach, we then targeted two putative HIF2A binding sites within this region. This resulted in ~ 40% reduction in *KLF6* expression (Fig. 4f), an effect of a similar magnitude to that caused by VHL reintroduction (Fig. 4b). Collectively, these data suggest that HIF2A supports *KLF6* expression by acting through the large *KLF6* super

enhancer, potentially explaining the relatively high *KLF6* levels in ccRCC when compared to other tumour types (Supplementary Fig. 1a).

We then performed transcriptomic analysis by RNA-seq on *KLF6* CRISPRi and control 786-M1A cells. As expected, the most significantly downregulated gene in the *KLF6*-targeted cells was *KLF6* itself (Fig. 5a and Supplementary Data 1). In agreement with the finding that KLF6 operated at least partially downstream of the HIF2A pathway, gene set enrichment analysis on the differentially expressed genes revealed a highly significant association between *KLF6*-depletion and downregulation of the canonical hypoxia-response gene set (Supplementary Fig. 6a). To study this further, we expressed exogenous KLF6 in the VHL-reintroduced 786-M1A cells and assessed the expression of the HIF2A downstream targets *CCND1, VEGFA* and *BHLHE40*. As expected, the expression of these genes was downregulated in the VHL-reintroduced cells. Reintroduction of KLF6 restored *BHLHE40* expression whereas *CCND1* and *VEGFA* expression remained low (Supplementary Fig. 6b). KLF6-dependent reversal of the negative effect of VHL reintroduction on *BHLHE40* expression was also observed in OS-LM1 cells (Supplementary Fig. 6c). In agreement, *BHLHE40* expression was attenuated in *KLF6*-depleted cells and KLF6 reintroduction was able to restore *BHLHE40* expression (Supplementary Fig. 6d–e). Thus, a part of

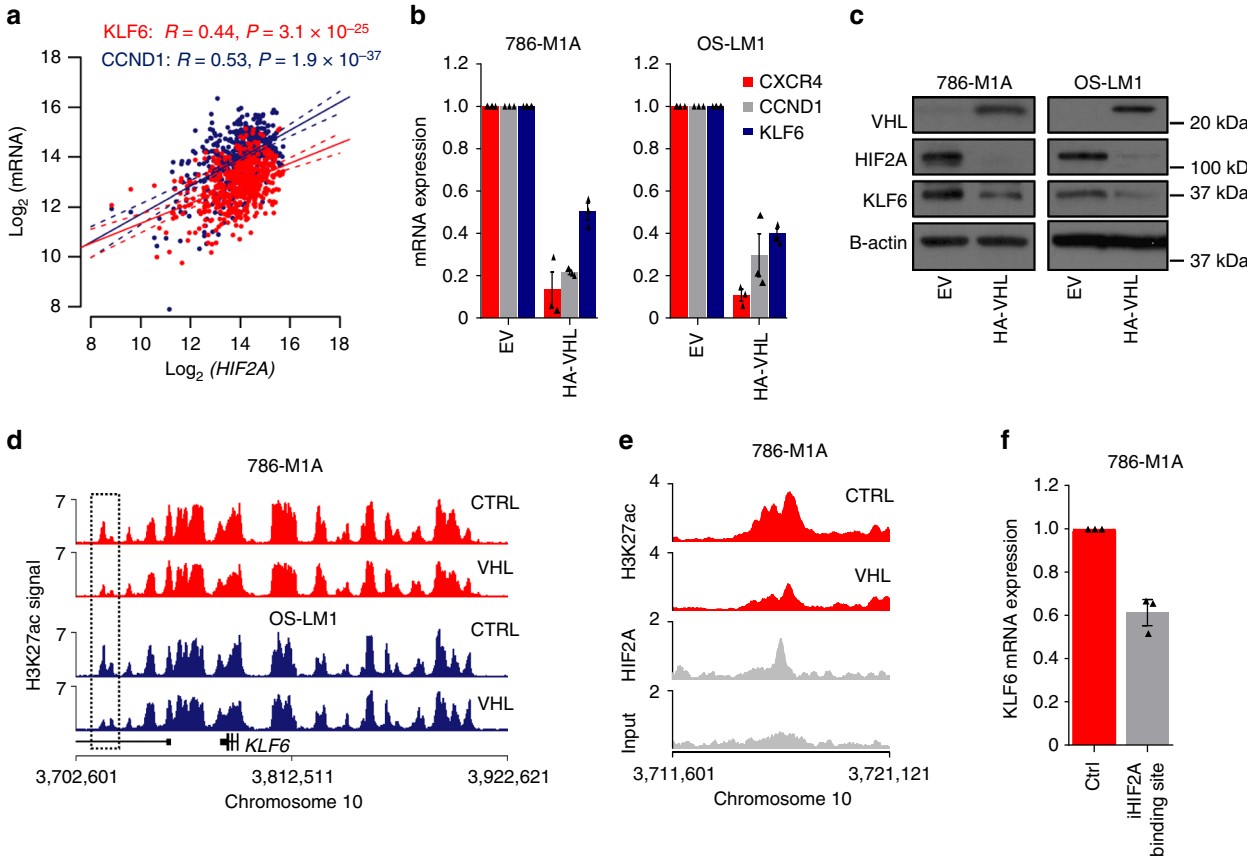

**Fig. 4** HIF2A modulates *KLF6* expression in ccRCC. **a** *KLF6* (red) and *CCND1* (blue) expression correlates with *HIF2A* expression in the clinical TCGA ccRCC data set. Pearson's correlation coefficient. **b** Expression of *KLF6* and the HIF2A downstream targets, *CXCR4* and *CCND1*, as measured by qRT-PCR, in the HA-VHL reintroduced 786-M1A and OS-LM1 cells. Average of three experiments. Error bars, SEM. **c** KLF6, HIF2A and VHL immunoblot of cells transduced with HA-VHL or empty vector. Representative of three experiments for 786-M1A and two for OS-LM1 cells. **d** H3K27ac ChIP-seq signal of 786-M1A and OS-LM1 cells transduced with empty vector or HA-VHL. A region significantly altered in both HA-VHL reintroduced cells when compared to the empty vector control highlighted by the dashed box. **e** A close up of the region highlighted in **d** for 786-M1A cells together with HIF2A ChIP-seq signal in the parental 786-M1A cells. **f** KLF6 mRNA expression level in a population of 786-M1A cells in which two putative HIF2A binding sites were targeted using CRISPRi. Average of three experiments. Error bars, SEM

the HIF2A transcriptional programme in ccRCC is also supported by its downstream transcriptional effector KLF6. Even though HIF2A regulates *KLF6* expression, it remains possible that the downregulation in *BHLHE40* expression upon VHL restoration is not dependent on reduced KLF6 expression; the data are also compatible with a model in which KLF6 and HIF2A regulate *BHLHE40* expression independently. However, the data show functional interaction between the HIF2A and KLF6 pathways. This observation is similar to that described for *PBRM1*, the loss of which has been linked to alterations in the transcriptomic consequences of *VHL* loss in ccRCC[36]. Multiple independent pro-tumorigenic factors thus seem to modulate the downstream effects of the VHL-HIF2A tumour driver pathway in ccRCC.

**KLF6 maintains lipid homeostasis in ccRCC.** Despite KLF6 being a component of the HIF2A pathway, the phenotype of KLF6-depleted cells was not similar to VHL-reintroduced ccRCC cells, which normally do not show a proliferative defect under standard tissue culture conditions[4]. Hence, we performed an unbiased pathway analysis on the RNA-seq data to identify potential KLF6 downstream effectors that could explain the reduced fitness of the KLF6-depleted cells. Pathways related to lipid homeostasis such as cholesterol and triacylglycerol bio-synthesis were among the topmost significantly altered pathways

with several key genes being downregulated in the *KLF6* knock-down cells (Fig. 5b). Gene set enrichment analysis also revealed a significant association between genes downregulated upon KLF6 inhibition and those involved in cholesterol homeostasis (Fig. 5c). A targeted analysis of the key genes that participate in each step of triacylglycerol and cholesterol biosynthesis pathways revealed that many of these genes were downregulated in the *KLF6* knockdown cells (Supplementary Fig. 7a). In addition, two critical transcription factors, *SREBF1* and *SREBF2*, that support the expression of lipid homeostasis genes, were also downregulated by KLF6 inhibition. Moreover, analysis of published ChIP-seq data sets[37,38] revealed that the regulatory regions of many of these genes were directly bound by SREBF1 and/or SREBF2 (Supple-mentary Fig. 7a). qRT-PCR analyses validated the reduced expression of *SREBF1* and *SREBF2* as well as *SCD* and *LSS*, two enzymes involved in the key intermediate steps of triacylglycerol and cholesterol biosynthesis pathways, respectively (Supplemen-tary Fig. 7b). Furthermore, we observed a reduction in the total cholesterol level in the KLF6-depleted cells (Fig. 5d). To test the contribution of lipid homeostasis perturbation on the phenotype of KLF6 inhibited cells, we combinatorially inhibited *SREBF1* and *SREBF2* expression in 786-M1A cells by CRISPRi (Supplementary Fig. 8a). In line with the effects of KLF6 inhibition, this led to a reduction in total cellular cholesterol levels (Fig. 5e) and impaired cell proliferation (Fig. 5f). Moreover, treatment with fatostatin, a

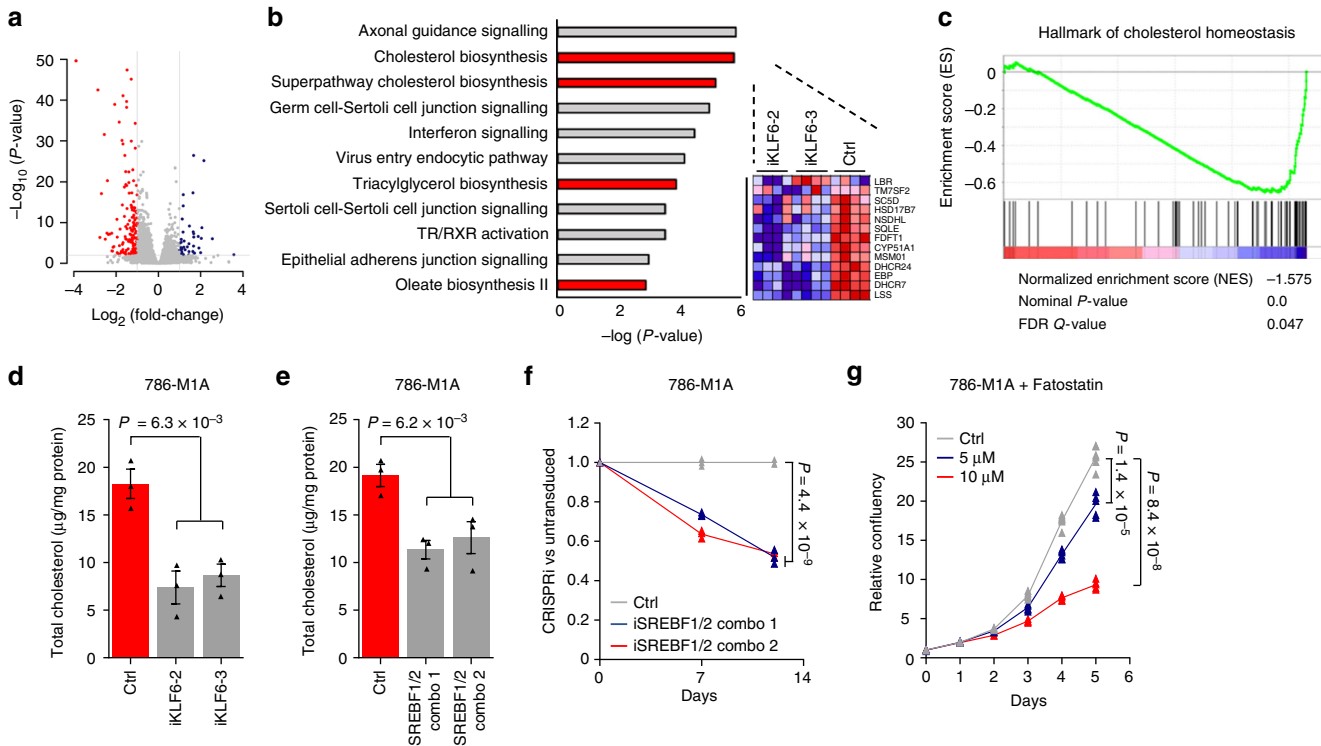

**Fig. 5** KLF6-dependent lipid homeostasis pathways promote ccRCC growth. **a** Volcano plot showing the differentially expressed genes in KLF6-targeted versus control 786-M1A cells (lentivirally delivered CRISPRi). Blue and red circles represent genes that were significantly upregulated and downregulated, respectively, in KLF6-targeted cells. **b** Most significantly deregulated pathways upon KLF6 inhibition as determined by Ingenuity Pathway Analysis. Highlighted in red are pathways involved in lipid homeostasis. Heatmap (right) shows significant downregulation of several key genes in the cholesterol biosynthesis pathway. Blue, low expression; red, high expression. **c** Gene set enrichment analysis shows downregulation of cholesterol homeostasis-related genes in KLF6-depleted cells. Blue, low correlation; red, high correlation. **d** Total cholesterol level in KLF6-targeted 786-M1A CRISPRi cells. Average of three experiments. Error bars, SEM. Two-tailed Student's t-test. **e** Total cholesterol level in SREBF1 and SREBF2-targeted 786-M1A CRISPRi cells. Average of three experiments. Error bars, SEM. Two-tailed Student's t-test. **f** Competitive proliferation assay of SREBF1 and SREBF2-targeted 786-M1A CRISPRi cells. The relative fraction of iSREBF1/2 double targeted cells and NTC transduced cells, normalized to day 0, compared to untransduced cells. Average of three technical replicates. Two-tailed Student's t-test. **g** Proliferation of 786-M1A cells treated with either vehicle or indicated concentrations of fatostatin. Average of six technical replicates. Two-tailed Student's t-test

SREBF1 and SREBF2 inhibitor[39], also reduced the proliferation of 786-M1A cells in a dose-dependent manner (Fig. 5g). Similar effects were observed with simvastatin, an inhibitor of HMG-CoA reductase, a critical enzyme in the cholesterol biosynthetic pathway (Supplementary Fig. 8b). Collectively, these data suggest that KLF6 promotes ccRCC fitness by supporting the expression of lipid metabolism effectors.

**Co-regulation of lipid homeostasis by KLF6 and mTORC1.** The mTORC1 complex modulates lipid biosynthesis via regulation of SREBF1 and SREBF2 expression and activity[40–42]. Thus, our observation that KLF6 regulated SREBF1 and SREBF2 expression, but also the expression of their downstream targets, was compatible with at least two mutually non-exclusive possibilities. First, KLF6 could positively regulate SREBF1 and SREBF2 mRNA expression, which would have secondary effects on their downstream targets. Alternatively, KLF6 could positively regulate mTOR activity, which would translate into increased SREBF1 and SREBF2 levels and activity. To distinguish between these two possibilities, we first tested whether mTOR regulated SREBF1 and SREBF2 in our systems. As expected, the mTOR antagonist torin 1 strongly inhibited mTORC1 activity, as indicated by the reduction of the phosphorylated forms of p70 S6 kinase (P-p70 S6 kinase) and ribosomal S6 (P-S6) (Supplementary Fig. 9a). This also led to decreased expression of the active forms of SREBF1

and SREBF2 (Supplementary Fig. 9a), and reduced total cholesterol (Supplementary Fig. 9b). Xenograft analysis showed that the mTOR pathway activity supported ccRCC growth in vivo (Supplementary Fig. 9c). In line with reduced SREBF1 and SREBF2 activity, the expression of SCD and LSS was reduced in torin 1-treated cells. Additionally, we observed a reduction in SREBF1 mRNA (Supplementary Fig. 9d). However, in contrast to the effects seen upon KLF6 inhibition (Supplementary Fig. 7b), torin 1 treatment led to an increase in SREBF2 mRNA expression (Supplementary Fig. 9d). Thus, while mTOR inhibition phenocopied the effects of KLF6 depletion at the level of SREBF1 and SREBF2 downstream targets, KLF6 inhibition additionally resulted in a reduction of SREBF2 mRNA expression, suggesting possible direct effects of KLF6.

To test the role of KLF6 as a direct regulator of lipid metabolism genes, we then performed ChIP-seq on flag-tagged KLF6 that was introduced into KLF6-depleted cells. qRT-PCR analysis confirmed that the tagged KLF6 was functional and it could rescue the expression of a lipid metabolism gene (Supplementary Fig. 10a). We identified >11,000 KLF6 peaks, distributed across different gene regulatory regions, most prominently at promoters, with an enrichment for a motif bound by members of the KLF family (Supplementary Fig. 10b). Importantly, the proximal regulatory regions of SREBF1, SREBF2, SCD, and LSS contained KLF6 peaks (Supplementary Fig. 10c–f). However, it still remained possible that mTOR signalling was also

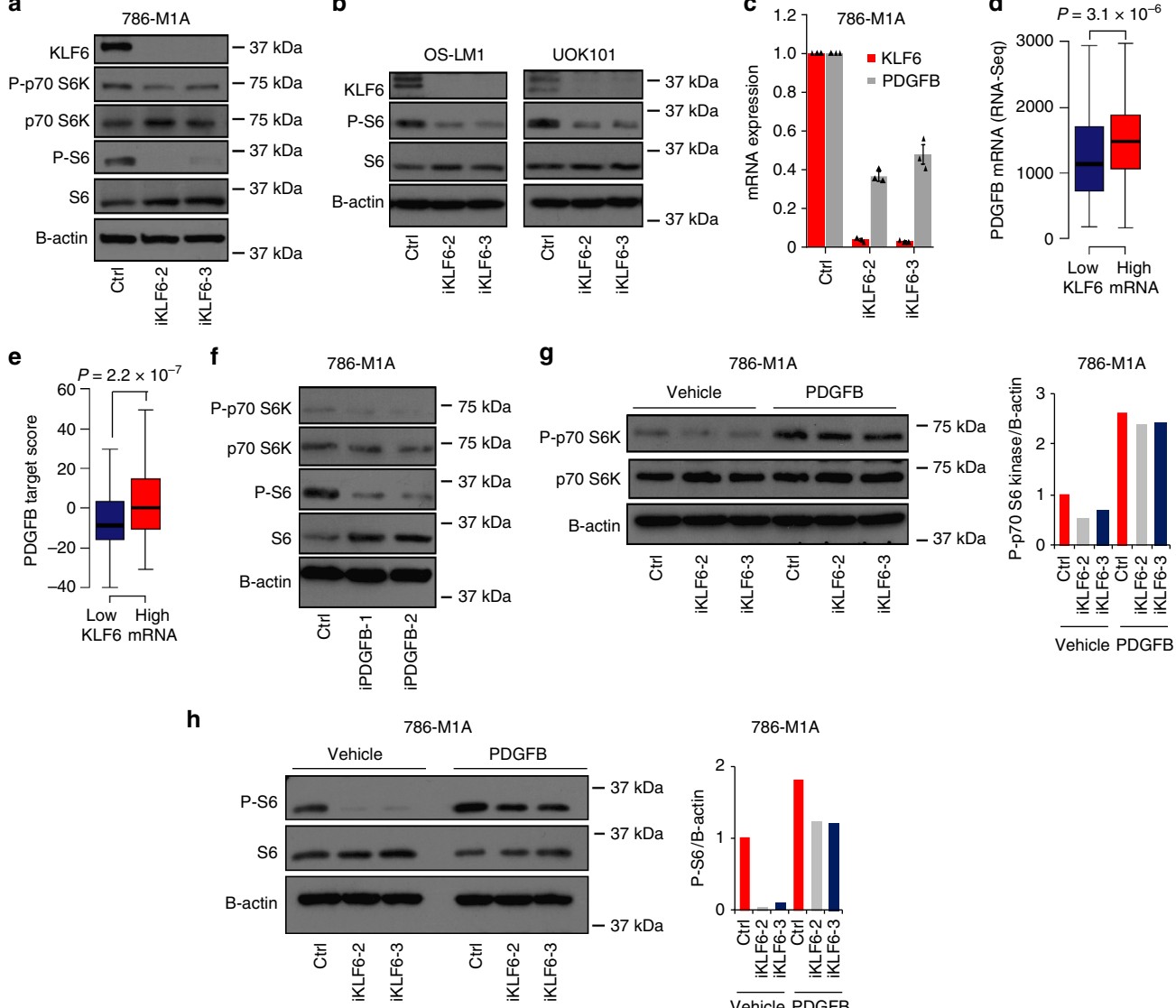

**Fig. 6** KLF6 modulates mTORC1 activity via transcriptional regulation of *PDGFB* expression. **a** mTORC1 activity in KLF6-targeted 786-M1A CRISPRi cells as determined by immunoblotting. Representative of three experiments. **b** mTORC1 activity in the pool of KLF6-targeted OS-LM1 and UOK101 CRISPRi cells. Representative of two and three experiments are shown for OS-LM1 and UOK101 cells, respectively. **c** *PDGFB* expression in KLF6-targeted 786-M1A CRISPRi cells as measured by qRT-PCR. Average of three experiments. Error bars, SEM. **d** *PDGFB* expression (RSEM normalized counts) in clinical ccRCC samples with either high (top 50%) or low (bottom 50%) *KLF6* expression. TCGA ccRCC cohort. Mann–Whitney *U*-test. Boxplot represents median and 25th and 75th percentiles, whiskers 1.5 times the interquartile range. **e** PDGFB target gene expression (sum of Z-scores) in clinical ccRCC samples with either high (top 50%) or low (bottom 50%) *KLF6* expression. TCGA ccRCC cohort. Mann–Whitney *U*-test. Boxplot represents median and 25th and 75th percentiles, whiskers 1.5 times the interquartile range. **f** mTORC1 activity in PDGFB-targeted 786-M1A CRISPRi cells as determined by immunoblotting. Representative of two experiments. **g**, **h** mTORC1 activity in KLF6-targeted 786-M1A CRISPRi cells supplemented with either human recombinant PDGFB (10 ng/ml) or vehicle control for 1 h. Representative of two experiments. Quantification of immunoblot bands shown on the right

contributing to the lipid metabolic phenotype of KLF6-depleted cells. In line with this possibility, mTORC1 activity was reduced in *KLF6*-inhibited cells (Fig. 6a, b). Collectively, these data demonstrate that KLF6 supports lipid homeostasis at two levels. First, KLF6 directly regulates the expression of lipid metabolism genes. Second, KLF6 promotes mTOR signalling which enhances the activation of lipid metabolism through SREBF1 and SREBF2.

**KLF6 supports mTOR activity via enhanced *PDGFB* expression.** Given that KLF6 is a transcription factor with no previously characterized link to the mTOR pathway, we investigated whether KLF6 transcriptionally activated a known mTOR agonist. RNA-

seq data for downregulated genes in the KLF6-depleted cells highlighted *platelet derived growth factor subunit B* (*PDGFB*) as a possible candidate. PDGFB can activate the mTOR pathway and previous data suggests that it also modulates SREBF1 and SREBF2 activity[43]. The qRT-PCR analysis confirmed that *PDGFB* was downregulated in KLF6-targeted cells (Fig. 6c and Supplementary Fig. 11a–b) and reintroduction of KLF6 into *KLF6* knockdown cells was able to rescue *PDGFB* expression (Supplementary Fig. 12a–b). Analysis of KLF6 ChIP-seq data showed that KLF6 binds to the proximal regulatory regions of *PDGFB*, a finding confirmed by ChIP-qPCR analysis (Supplementary Fig. 12c–d). Supporting the clinical relevance of our finding, ccRCC patient samples with high level of *KLF6* expression also

expressed higher levels of *PDGFB* in the TCGA data set (Fig. 6d). Furthermore, gene set enrichment analysis demonstrated a significant downregulation of a previously derived PDGFB-responsive gene signature[43] in KLF6-depleted cells (Supplementary Fig. 12e). The PDGFB response signature was also more active in patient samples with higher *KLF6* expression in the ccRCC TCGA data set (Fig. 6e). Finally, to directly test the role of PDGFB as a regulator of mTOR in our systems, we targeted PDGFB by CRISPRi (Supplementary Fig. 12f) and assessed the levels of P-p70 S6 kinase and P-S6. Knocking down *PDGFB* inhibited mTORC1 activity (Fig. 6f), phenocopying the effects of KLF6 inhibition (Fig. 6a). Conversely, supplementing *KLF6*-targeted cells with recombinant human PDGFB re-activated mTOR signalling (Fig. 6g, h). This suggested that KLF6 supports mTOR signalling by directly regulating the expression of PDGFB.

## Discussion

We report the identification of a cellular signalling loop that links the transcription factor KLF6 to lipid metabolic activity, enhanced tumour growth and metastatic colonization in ccRCC. *KLF6* expression is supported by a large super enhancer that is partially activated by the ccRCC-initiating VHL-HIF2A pathway. The high prevalence of *VHL* mutations and consequent HIF2A stabilization could thus explain the relatively high KLF6 mRNA levels observed in human ccRCCs. Transcriptional profiling revealed a prominent lipid metabolism defect in KLF6-targeted ccRCC cells. We find that KLF6 has a dual effect on lipid homeostasis as a regulator of both the transcription of lipid metabolism genes, and the activation of SREBF1 and SREBF2 through the PDGFB-mTOR axis (Fig. 7). These results suggest a mechanism for the prevalent mTOR activation in human ccRCC. Moreover, the links between super enhancer-driven transcriptional networks and the activity of essential metabolic pathways may explain the stability of cell identity-defining transcriptional programmes in cancer.

Previous reports have suggested that large cancer-associated enhancer clusters or super enhancers may be particularly sensitive to perturbations, making them putative therapeutic targets[21]. We characterized one of the strongest super enhancer loci in ccRCC cells by targeted inhibition of several of its constituent enhancers using CRISPRi and by deleting a large 113 kb segment of the super enhancer using CRISPR-Cas9 gene editing. Only some of the constituent enhancers were individually essential for full *KLF6* expression. Simultaneous targeting of two enhancers more strongly reduced *KLF6* expression, but even then the effects were relatively subtle. Further reduction in *KLF6* was observed when all five enhancers were inactivated simultaneously. Finally, even in the cells with a large deletion that covered several individual enhancers, KLF6 mRNA level was reduced by only ~65%. *KLF6* expression is thus regulated by a robust super enhancer which is insensitive to alterations in the activity of many of its constituent enhancers. Such redundancy is well-aligned with the fact that most biological processes and developmental transcriptional programmes are insensitive to environmental and other incoming variation[44]. It is interesting to note, however, that VHL restoration and the consequent inactivation of HIF2A was able to reduce *KLF6* expression by ~50%. Taken together, our analysis of the upstream regulation of *KLF6* expression suggests that the identification of relevant gene regulatory elements simply based on chromatin profiling is challenging[45] and that functional analysis of enhancer function is needed. Also, therapeutic targeting of transcriptional addictions in cancer may at least in some contexts be more complicated than what has been suggested by some reports[24].

KLF6 has been described to have both growth suppressive and supportive functions in different cancers[46–48]. Large-scale cancer genome re-sequencing efforts suggest that *KLF6* is rarely inactivated via genetic alterations[49,50]. In ccRCC specifically, the large TCGA cohort of 448 cases reports no point mutations and only one tumour with a deep deletion of *KLF6*. In this particular case, *KLF6* is lost together with several other genes on chromosome 10p, suggesting that the alteration is not specifically targeting *KLF6*. On the other hand, the expression and chromatin alteration patterns we observe in human ccRCCs and cell lines are supportive of a pro-tumorigenic role for KLF6. In contrast to breast cancer where a *KLF6* splice variant promotes metastatic progression[47], our results suggest that full-length KLF6, downstream of a robust super enhancer, promotes ccRCC growth in vitro and in vivo. Mechanistically, we demonstrate that KLF6

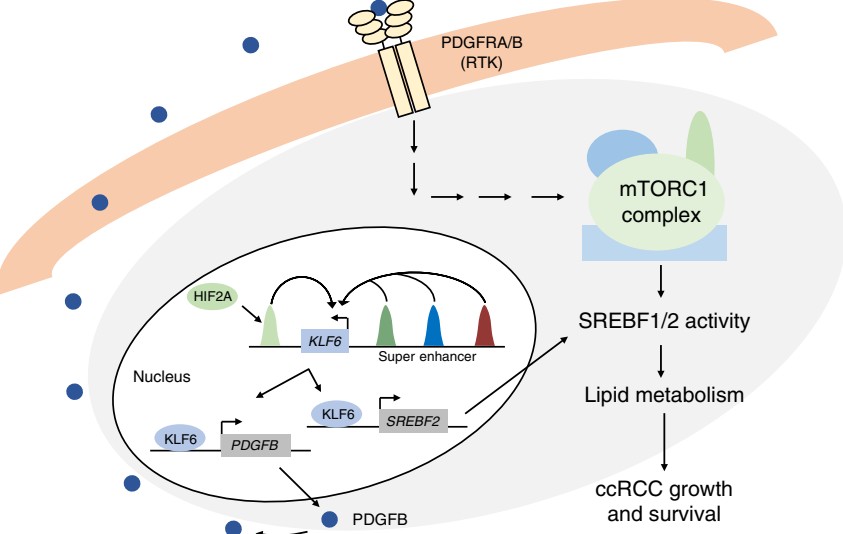

**Fig. 7** Model: A robust super enhancer, partially activated by HIF2A, supports *KLF6* expression in ccRCC. KLF6 promotes the expression of *PDGFB*, leading to the activation of mTORC1 and the downstream regulators of lipid metabolism, SREBF1 and SREBF2. Additionally, KLF6 supports the expression of *SREBF2*, and possibly also *SREBF1* and their downstream targets, independently of mTORC1. Collectively these two pathways enhance lipid metabolism, leading to increased ccRCC growth

regulates the expression of lipid homeostasis genes directly, but it also enhances the activation of SREBF1 and SREBF2 by supporting the expression of *PDGFB*, a prominent activator of the mTOR pathway. Genetic and pharmacological inhibition of SREBF1 and SREBF2 activity also reduced ccRCC proliferation, suggesting that effects on lipid metabolism explain at least partially the phenotypic consequences of KLF6 depletion, although our results do not exclude the possibility that KLF6 promotes ccRCC growth through other mechanisms as well. However, we do not understand the reasons for the differences between our results and those of Gao et al. who have recently reported a tumour suppressive role for KLF6 in ccRCC[46]. Gao et al. also report that KLF6 represses the expression of *E2F1*, an effect we do not observe (Supplementary Fig. 13a).

Two genes, *CPT1A* and *PLIN2*, have recently been implicated in HIF-dependent lipid accumulation in ccRCC[51,52]. *CPT1A* is repressed by HIF1A and HIF2A, consequently leading to lipid droplet formation and enhanced ccRCC growth due to reduced fatty acid transport into the mitochondria. *PLIN2*, encoding a lipid droplet coat protein, is positively regulated by HIF2A, and it promotes lipid droplet accumulation and ccRCC fitness. Based on our RNA-seq data, the expression of both *CPT1A* and *PLIN2* is downregulated upon KLF6 depletion. This suggests that in addition to regulating the SREBF1 and SREBF2 pathways, KLF6-dependent effects on ccRCC lipid metabolism may also involve PLIN2 and CPT1A. However, as VHL reintroduction increases CPT1A expression but decreases *KLF6* expression, and KLF6 supports CPT1A expression (Supplementary Fig. 13b), it seems unlikely that the pro-tumorigenic effects of KLF6 are directly caused by modulation of CPT1A activity through the mechanism recently described by Du et al.[51]. Further work is thus needed for a comprehensive understanding of how different mediators of lipid metabolic phenotypes contribute to ccRCC.

The mTOR pathway can be activated by multiple different signals[53]. We find that in ccRCC cells, mTOR activity is supported by PDGFB. Given that the 786-M1A cells carry a homozygous truncating mutation in *PTEN*[31], our data suggest that upstream PDGFR activation is required for mTOR activation in ccRCC cells even when *PTEN* is lost. Interestingly, our results functionally link two approved therapeutic targets in ccRCC, the PDGFR and mTOR pathways, suggesting a molecular explanation for the favourable effects of an mTOR/PDGFR combinatorial treatment approach in ccRCC[54]. In a phase II randomised trial of everolimus and lenvatinib (an inhibitor of both PDGFRA and PDGFRB[55]) in combination versus the drugs as monotherapy for patients with metastatic renal cell carcinoma who have progressed after one previous VEGF-targeted therapy, lenvatinib plus everolimus were found to act synergistically to provide a progression-free survival (PFS) benefit over either drug alone, although a significant difference was seen only over everolimus (median PFS 14·6 months vs 5·5 months)[54]. However, although grade 3 and 4 events occurred in fewer patients allocated single-agent everolimus (50%) compared to lenvatinib plus everolimus (71%), there was greater toxicity in those patients assigned lenvatinib alone (79%); overall, the toxicity was manageable with dose reductions.

In conclusion, we describe molecular signalling network that links the super enhancer-associated transcription factor KLF6 to lipid metabolism and enhanced mTOR activity in ccRCC. Our experimental analyses of the super enhancer upstream of *KLF6* demonstrate the robustness of some cancer-associated super enhancers. This may complicate the usage of therapeutic agents that aim at super enhancer inactivation in cancer. Finally, the link between super enhancer-associated programmes and core metabolic pathways may provide clues to the mechanisms that maintain cell identity-defining transcriptional networks in cancer.

## Methods

**Cell lines and reagents**. The human RCC cell lines 786-O, 786-M1A, OS-RC2, OS-LM1, RCC-MF were obtained from J. Massagué (MSKCC, New York, NY) in 2014. The UOK101 cell line was obtained from Marston Linehan (the UOB Tumor Cell Line Repository, National Cancer Institute, Bethesda, MD) in 2014. 786-M1A and OS-LM1 are metastatic derivatives of 786-O and OS-RC2 cells, respectively[28]. The A549 lung cancer cell line was obtained from C. Martins (MRC Cancer Unit) in 2017. The identity of all cell lines was confirmed by STR analysis. All cells were confirmed to be mycoplasma negative by using the MycoAlert™ Mycoplasma Detection Kit (Lonza, LT07-318). At the time of the study, none of the cell lines in this paper were listed in the database of commonly misidentified cell lines maintained by ICLAC. All RCC cell lines were cultured in RPMI-1640 medium (Sigma) supplemented with 10% FBS, penicillin (100 U mL$^{-1}$) and streptomycin (µg mL$^{-1}$). For lentivirus production, HEK293T cells were used and cultured in DMEM (Invitrogen), supplemented with 10% FBS, penicillin (100 U mL$^{-1}$) and streptomycin (µg mL$^{-1}$).

Doxycycline-inducible Cas9, pCW-Cas9 was a gift from Eric Lander and David Sabatini (Addgene plasmid #50661)[56]. LentiCas9-Blast was a gift from Feng Zhang (Addgene plasmid #52962)[57]. pHR-SFFV-KRAB-dCas9-P2A-mCherry was a gift from Jonathan Weissman (Addgene plasmid #60954)[58]. pKLV-U6-gRNA(BbsI)-PGKpuro2ABFP, was a gift from Kosuke Yusa (Addgene plasmid #50946)[59]. This vector was modified to generate the following plasmid variants: (i) pKLV-U6-gRNA(BbsI)-PGKhygro2ABFP, (ii) pKLV-U6-gRNA(BbsI)-PGKhygro2AmCherry (iii) pKLV-U6-gRNA(BbsI)-PGKhygro2AeGFP. pLVX-Puro (Clontech #632164) was used to express the exogenous cDNA constructs. For lentivirus production, packaging plasmids, psPAX2 (Addgene plasmid #12260) and pMD2.G (Addgene plasmid #122259) were gifts from Didier Trono. All sgRNA constructs and primers used in this study were purchased from Sigma-Aldrich and the sgRNA sequences are available in Supplementary Table 1 and 2. Everolimus was purchased from APExBIO (A8169). Fatostatin (F8932), simvastatin (S6196), propylene glycol (W294004) and Tween-80 (P4780) were purchased from Sigma-Aldrich.

**Lentiviral transduction**. HEK293T cells were transfected with a mixture of the lentiviral transfer plasmid of interest, psPAX2 and pMD2.G using FuGENE 6 transfection reagent (Promega E269A). Media containing the lentivirus was collected 48–72 h post-transfection and filtered through a 0.45 µM PVDG sterile filter. Cells were transduced with the lentiviral supernatant in the presence of 8 µg/mL Polybrene (Millipore).

**Chromatin immunoprecipitation**. Sub-confluent cells were trypsinized and crosslinked with 1% formaldehyde-supplemented media for 10 min. The reaction was quenched with 0.125 M glycine for 5 min and washed with PBS twice. The cells were either pelleted and stored at −80 ºC, or subjected to immunoprecipitation. For IP, protein A/G magnetic beads (Thermo, 26162) were first equilibrated by washing the beads with 0.5% BSA in PBS three times. The beads were then incubated with antibodies in 0.5% BSA in PBS at 4 ºC while rotating for a minimum of 4 h. The following antibodies were used: H3K27ac (Abcam, ab4729, 5 µg), monoclonal FLAG (Sigma-Aldrich, F1804, 30 µg), HIF2A (Novus Biologicals, NB100-122, 30 µg) and rabbit polyclonal IgG (Abcam, ab27478, 5 µg). The cross-linked cells were resuspended and dounced in lysis buffer (20 mM Tris-HCl pH 8.0, 150 mM NaCl, 2 mM EDTA pH 8.0, 0.1% SDS and 1% Triton X-100), followed by sonication using the Bioruptor (Diagenode) for 14 cycles, 30" on/30" off. The lysates were spun down at 4 ºC for 20 min at 14,000 rpm. The supernatants were added onto the antibody-conjugated magnetic beads and incubated overnight at 4 ºC while rotating. On the following day, the beads were washed three times with low salt buffer (50 mM HEPES pH 7.5, 140 mM NaCl, 1% Triton) and once with high salt buffer (50 mM HEPES pH 7.5, 500 mM NaCl, 1% Triton). DNA bound to the antibody-conjugated beads was eluted with elution buffer (50 mM NaHCO₃, 1% SDS) and de-crosslinked by shaking at 1000 rpm for 3 h at 65 ºC. De-crosslinked DNA was purified using the QuickClean II PCR Extraction Kit (Genescript L00419-100) according to the manufacturer's recommendations.

**ChIP-seq library preparation**. Purified ChIP DNA was subjected to Illumina sequencing. Sequencing libraries were prepared using the KAPA Hyper Prep Kit (KR0961) according to the manufacturer's recommendations. Adapter-ligated libraries were size-selected using Agencourt AmPure XP beads (Beckman Coulter, A63880) to obtain fragments of 150–350 bp. Size-selected fragments were amplified for 15 cycles using the KAPA HiFi HotStart Ready mix and the amplified libraries were pooled in equimolar concentration for sequencing.

**RNA-seq**. Total RNA was extracted from sub-confluent cells in four replicates using the RNeasy Mini Kit (Qiagen 74104) according to the manufacturer's protocols. RNA concentration and quality were assessed with the Agilent RNA Nano 6000 kit (Agilent 5067-1511) on Agilent Bioanalyzer 2100 instrument. RNA-seq libraries were prepared using the SENSE mRNA-Seq Library Prep Kit V2 (Lexogen) following the manufacturer's recommendations with 1 µg of total RNA as the starting material. The size and quality of the final library products were assessed using the Agilent High Sensitivity DNA Kit (Agilent 5067-4626). Library concentration was determined using the KAPA Library Quantification Kit (KR0405).

Libraries were pooled in equimolar concentrations and subjected to Illumina sequencing.

**ChIP-seq and RNA-seq data analysis**. ChIP-seq and RNA-seq sequencing were performed on Illumina HiSeq HiSeq 4000 systems using 50 bp single-end runs. Raw ChIP-seq sequencing reads were aligned to hg38 using *bowtie2*[60] and the resulting sam files were converted into sorted bam files using *samtools*[61]. Peaks were called using MACS2[62]. RNA-seq sequencing reads were aligned to hg38 using RSEM[63] and bowtie2 with default settings. Gene set enrichment analysis (GSEA), a computational method that determines whether an 'a priori' defined set of genes demonstrates statistically significant, concordant differences between two biological states, was performed using the gene expression data[64]. The GSEA software (version 2-2.2.2) and Molecular Signature Database (MSigDB) (version 6.0) (http://www.broad.mit.edu/gsea/) were used together with gene lists ranked by normalized RNA-seq counts (CPMs). Normalized enrichment score (NES) and false discovery rate (FDR) ($p > = 0.25$) were computed and used to quantify the magnitude of enrichment and statistical significance, respectively. Pathway analyses of RNA-seq data were performed with Ingenuity Pathway Analysis (IPA) software. (QIAGEN Inc., https://www.qiagenbioinformatics.com/products/ingenuitypathway-analysis). The fatty acid and cholesterol biosynthesis pathways were assembled using information integration from a number of pathway databases (KEGG, Reactome, Wikipathways, and Ingenuity Pathway Analysis) and pathway maps[65–67]. SREBF1/2 targets were identified using the TRRUST database[37] and the Harmonizome resource[38]. Publicly available SREBF1/2 ChIP-seq data sets were collected from NCBI GEO and SRA databases (SRP007993, SRP028819, SRP097662, SRP012412). Fastq files were aligned using Bowtie2 (version 2.3.4.3) to the hg38 reference genome from UCSC genome database and peaks were called using MACS2 (version 2.1.0). Peaks from a consensus peak set (peaks detected in at least two samples) were annotated to genes if they were maximum 2.5 kb away from the gene promoter, or on the gene body and had overlaps with TSSs identified by the FANTOM project[68] using the ChIPpeakAnno R/Bioconductor package[69].

**Human samples**. The ccRCC tumour samples for ChIP-seq analyses have been previously reported[27]. They were collected from nephrectomy specimens in accordance with the Declaration of Helsinki. Patients were recruited for tissue donation by providing written consent and after study approval from an Institution Review Board (NRES Committee East of England-Cambridge Central, NHS Health Research Authority; REC 03/018).

**Animal studies**. All animal experiments were performed in accordance with protocols approved by the Home Office (UK) and the University of Cambridge Animal Welfare and Ethical Review Body (PPL 70/7990). For subcutaneous tumour growth assay, cells were pelleted and resuspended in PBS/Matrigel Matrix (BD) mix at 1:1 ratio. $1 \times 10^5$ cells in 100 μL solution were injected into each flank of 5–8 weeks old athymic male nude mice (Charles River Laboratories). Tumour growth was monitored by IVIS bioluminescence imaging (Perkin Elmer) and calliper measurement. Tumour volume (V) was calculated using the equation V = (length × width$^2$) × 0.5. For the lung colonization assay, cells were pelleted and resuspended in 1× sterile PBS. $3 \times 10^5$ cells in 100 μL solution were injected into the lateral tail vein of 5–7 weeks old NOD/SCID mice (Charles River Laboratories), followed by bioluminescence imaging. At the end of the assay, lungs were extracted and processed for immunohistochemistry. For the in vivo everolimus experiment, 500,000 cells were subcutaneously injected into 6 week-old athymic nude male mice. After the tumours became palpable, mice were separated into two groups with equal average tumour size and treated orally with either everolimus (5 mg/kg/daily) or vehicle (30% propylene glycol and 5% Tween-80 in sterile water) for 3 weeks. Tumour growth was monitored by calliper measurement.

**Histology and immunohistochemistry**. Lungs were harvested from euthanized mice and fixed in 10% formalin overnight. The formalin-fixed samples were sent to the Cambridge University Hospital Human Research Tissue Bank for paraffin-embedding and sectioning. Human vimentin (Cell Signaling Technology; Cat. 5741, 1:100) staining was performed in a Bond-Max instrument (Leica) using Bond Polymer Refine Detection reagents (Leica) according to the manufacturer's protocol (IHC Protocol F).

**In vitro competitive proliferation assay**. For CRISPR-Cas9 mutagenesis competitive proliferation assay, control and targeted cells which carried different fluorescent markers (BFP$^+$ or mCherry$^+$) were mixed at a 1:1 ratio and plated onto 6-wells plates. The percentage of each cell population was analysed at $T = 0$ and at multiple time points throughout the assays by flow cytometry on LSR Fortessa (BD Biosciences). The following gating approach was used: FSC-A, FSC-H, SSC-A to select for live and single cells, and then mCherry (561 nm/610 nm) and BFP (383 nm/445 nm) channels for discriminating between the two cell populations. Similar strategy was employed for the CRISPRi competitive proliferation assay with slight modification: the BFP$^+$ sgRNA-expressing/mCherry$^+$ CRISPRi cells were mixed with mCherry$^+$ only CRISPRi cells at a 1:1 ratio. The gating strategy for flow cytometry data analysis is shown in Supplementary Fig. 14.

**cDNA synthesis and quantitative real-time PCR**. Total RNA was extracted using the RNAzol®RT reagent (Sigma) according to the manufacturer's protocol. The concentration and quality of the RNA were determined by using the NanoDrop 1000 spectrophotometer (Thermo). RNA (500 ng) was converted into cDNA using the High-Capacity cDNA Reverse Transcription Kit (Thermo) according to the manufacturer's recommendations. cDNA was either stored at −20 °C or used for qRT-PCR gene expression analysis. qRT-PCR was performed using the 2× TaqMan Fast Advanced master mix (Thermo) and 20x pre-designed TaqMan gene expression probes (Thermo) on the StepOnePlus$^{TM}$ Real-Time PCR instrument (Thermo). The following TaqMan probes were used: KLF6 (Hs00810569_m1), EPAS1 (Hs01026149_m1), PDGFB (Hs00966522_m1), CXCR4 (Hs00607978_s1), CCND1 (Hs00765663_m1), VEGFA (Hs00900055_m1), BHLHE40 (Hs01041212_m1), SREBF1 (Hs01088679_g1), SREBF2 (Hs01081784_m1), SCD (Hs01682761_m1), LSS (Hs01552331_m1), E2F1 (Hs00153451_m1), CPT1A (Hs00912671_m1) and TBP (Hs00427620_m1). The Ct values of the gene of interest were normalized using the Ct value of the housekeeping control, TBP. The fold change of the gene expression was calculated using the $2^{-\Delta\Delta Ct}$ method.

**Protein extraction and western blotting**. Cells were either trypsinized or scraped on ice, washed once with 1× ice-cold PBS and lysed on ice in RIPA lysis buffer (Sigma-Aldrich) containing 1:100 1× protease inhibitor cocktail (Sigma-Aldrich) and 1:100 1× phosphatase inhibitor cocktail (Sigma-Aldrich). The protein lysates were quantified using the Pierce BCA Protein Assay Kit (Life Technologies) according to the manufacturer's protocol. Proteins were separated using SDS-PAGE gels and transferred onto PVDF membrane (Millipore). The membrane was blotted with primary antibody overnight at 4 °C. Antibodies used were KLF6 (Santa Cruz Biotech, sc-7158, 1:1000), HIF2A (Novus Biologicals, NB100-122, 1:1000), VHL (BD Biosciences, 564183, 1:1000), P-p70 S6-kinase (Cell Signaling Technology, Thr389, #9205, 1:1000), p70-S6-kinase (Cell Signaling Technology, #9202, 1:1000), P-S6 ribosomal (Cell Signaling Technology, Ser235/236, #4857, 1:3000), S6 ribosomal (Cell Signaling Technology, #2317, 1:1000), SREBP1 (Santa Cruz Biotech, sc-13551, 1:100), SREBP2 (Santa Cruz Biotech, sc-13552, 1:100), and B-actin (Sigma-Aldrich, A1978, 1:20000). Secondary antibodies were polyclonal goat anti-mouse IgG/HRP (Dako, P0447, 1:10000) and polyclonal goat anti-rabbit IgG/HRP conjugated (Dako, P0448, 1:5000). For the assessment of mTORC1 activity, cells were serum-starved overnight and protein was extracted followed by Western blotting. For the PDGFB supplementation experiment, cells starved overnight were treated with either 10 ng/mL recombinant human PDGF-BB (Peprotech #100-14B) or vehicle control for one hour. Full scans of Western blots are in Supplementary Fig. 15.

**Exogenous cDNA expression**. *KLF6* cDNA was amplified from 786-M1A cells and cloned into pLVX-Puro. Primers to amplify *KLF6* cDNA are listed in Supplementary Table 2. For the CRISPR-Cas9 competitive proliferation rescue assay, the sgRNA binding site of the exogenous KLF6 was synonymously mutated by site-directed mutagenesis.

**Total cholesterol quantification**. Lipids were extracted from cells using the chloroform-free lipid extraction kit (ab211044, Abcam) according to the manufacturer's recommendation. Briefly, cells were agitated in the extraction buffer for 20 min at room temperature, spun down and the supernatant was dried overnight at 37 °C. The lipid extract was resuspended in suspension buffer and sonicated for 20 min followed by agitation at 37 °C for 20 min. Total cholesterol was quantified using the Amplex Red Cholesterol Assay Kit (A12216, Invitrogen) according to the manufacturer's protocol. Total cholesterol was normalized to the total protein amount for each sample.

**Statistical analyses**. Statistical analyses were performed either in R or GraphPad Prism (Version 5). Sample sizes are denoted in figure legends. No statistical method was used to predetermine the sample size. P-values lower than 0.05 were considered statistically significant. For animal bioluminescence, tumour growth data and histological tumour count data a two-tailed Mann–Whitney U-test was used. No animals were excluded from the analyses. For histological tumour count analyses the experimental groups were blinded, for other experiments the experimental groups were not randomized or blinded. For parametric tests data normality was assessed using the Shapiro-Wilk test, variance between the groups was not assumed to be equal. No corrections for multiple testing were made. For competitive proliferation assays, cholesterol quantification and in vitro drug experiments data a two-tailed unpaired t-test with Welch approximation was used. For qRT-PCR, three independent experiments are shown unless stated otherwise in the figure legend, each of the experiment is the average of three technical replicates. Boxplots represent median and 25th and 75th percentiles, whiskers as indicated in figure legends.

**Reporting summary**. Further information on experimental design is available in the Nature Research Reporting Summary linked to this article.

**Code availability**. Custom code used in this study are available from the corresponding author on request.

## Data availability

Previously published ChIP-seq and RNA-seq data were re-analysed from the GEO data set GSE98015 and the SRA data sets SRP007993, SRP028819, SRP097662, SRP012412. The RNA-seq and ChIP-seq data generated during this study have been deposited to GEO under the access codes GSE115763 and GSE115749. Human RNA-seq data for different tumour types were downloaded from the TCGA data portal (http://tcga-data.nci.nih.gov/).

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

## Acknowledgements

We thank A. Speed for technical assistance with animal experiments, and R. Schulte, C. Cossetti and G. Grondys-Kotarba for cell sorting and assistance with FACS experiments. Infrastructure for the Cambridge Urological Bio-repository was funded by the Cambridge Biomedical Research Campus and CRUK Cambridge Centre. The Human Research Tissue Bank is supported by the NIHR Cambridge Biomedical Research Centre. We thank V. Gnanapragasam for administration of the DIAMOND study which allowed access to clinical RCC samples, and M. Linehan for the UOK101 cells. This work was supported by the Medical Research Council (MC_UU_12022/7 and MC_UU_12022/10). S.E.S. was supported by the Malaysian Ministry of Higher Education IPTA Academic Training fellowship.

## Author contributions

S.E.S. and S.V. designed the study. S.E.S. and S.V. prepared the manuscript with support from other authors. S.E.S., P.R., E.V., M.N.Z. and S.P. performed the experiments and acquired the data. J.B., A.Y.W. and G.D.S. provided clinical specimens. S.E.S., D.B., S.S., T.E. and S.V. analysed and interpreted the data. S.V. supervised the study.

## Additional information

**Competing interests:** T.E. is an employee of AstraZeneca on leave of absence from the University of Cambridge, and an author on the study reporting the results from the lenvatinib plus everolimus clinical trial. The remaining authors declare no competing interests.

