## [Peer Review File · Nature Communications]

Reviewers' comments:

Reviewer #1 (Remarks to the Author):

The manuscript by the Vanharanta lab identifies KLF6 expression as a HIF2A target highly upregulated in renal cell carcinoma cells. They go on and show that KLF6 expression promotes cancer cell proliferation and they propose that KLF6 acts by upregulating the mTOR/SREBP pathway through the transcriptional regulation of PDGFB. The data are clear and the techniques are sound. I just have few comments:

1- The effect on tumor volume after editing of KLF6 is minor (Fig. 2A and 2B). Please argue. Are the cells in vivo less dependent on PDGFB production? What's the status of the mTOR pathway? Are these tumors resistant to rapamycin?

2- The samples in Fig. 6F should be loaded on the same gel and quantified. What is the effect of PDGFB rescue on SREBP cleavage and activity?

Reviewer #2 (Remarks to the Author):

The manuscript from Syafruddin et. al. describes a transcription factor controlled signaling axis whereby KLF6 controls lipid homeostasis and tumor growth in clear cell Renal Cell Carcinoma through regulation of PDGFB and mTOR. The authors work helps to address how mTOR signaling is upregulated in many ccRCC tumors even without genetic or mutational alterations to mTOR signaling axis. Understanding this process could help lead to better targeted therapies for ccRCC patients. To address this and identify potential vulnerabilities of ccRCC cells, the authors profiled ccRCC cell lines and identified super enhancers that might regulate key pathways. The authors settled on analysis of KLF6 as the super enhancer regulating its expression was highly acetylated in ccRCC cells and KLF6 levels are upregulated in many ccRCC tumors. The authors characterize KLF6 function through KO and CRISPRi to determine that it is important in regulating cell and tumor growth in vitro and in vivo. Through profiling KLF6 depleted cells, the authors identify lipid metabolism and mTOR signaling as key regulated networks and determine that KLF6 up regulates PDGFB which in turn increases mTOR signaling. The authors have the foundations of an important model, but additional work is required to complete the loop.

The authors should complete the loop and demonstrate that PDGFB is a direct KLF6 target by ChIP. This is a key component to the author's model and it is important to know if PDGFB is a direct target or not. Further the authors should demonstrate that PDGFB is required in cells with high levels of KLF6 for maintaining mTOR signaling.

It is not clear how the lipid metabolism defects are connected to the PDGFB signaling loop. Does PDGFB supplementation restore the cholesterol defects of iKLF6 cells? Is this process mTOR regulated, does inhibiting mTOR block this? Are the cholesterol synthesis and lipid metabolism genes direct KLF6 targets or SREBF1/2 targets and are SREBF1/2 direct KLF6 targets. This information would help complete the signaling loop the authors are describing.

It is unclear how the super enhancer regulating KLF6 expression is functioning. Introduction of VHL downregulated KLF6 more strongly than any single super enhancer CRISPRi interference, but results in only modest change to a single upstream KLF6 super enhancer element that was not tested by CRISPRi. The authors should test the VHL/HIF2A regulated enhancer by CRISPRi and see if this results in more dramatic KLF6 suppression. While the authors argue that the super enhancer is a robust regulator that withstands interference, it may be that it is not the key or only regulator of KLF6 expression. If the authors combine more than 2 CRISPRi (maybe even all 5) without complete genetic deletion of the SE, does result in strong KLF6 suppression? Were other genes effected strongly by the deletion of the super enhancer (that aren't KLF6 targets)? More data is needed to ensure that super enhancer regulates KLF6 is needed.

The authors mention the clinical trials testing combos of mTOR and promiscuous VEGF/PDGFR inhibitors in ccRCC cancer and that combo is more effective than single agent therapy. Since these results are known, the authors should consider describing their results as a mechanistic reason to explain the effectiveness of the combination therapy rather than as a rationale for trying it. The authors need to better describe (in legends or text, or both) more about both the traditional CRISPR and CRISPRi cell lines. It is not clear anywhere if the authors selected individual clones or maintained the pools from initial infections.

Minor issues:

The authors should be careful about the use of the word autocrine since they provide no evidence that the PDGFB secreted acts on the same cell that secreted it.

Figure S1A/B- not clear what the y axis is, is this RPKM?

Figure S1C how about a non-ccRCC line with lower expression? How do we know these cells have elevated KLF6 expression when they are all the same?

Please be more clear if the CRISPR/Cas9 knockout clones in figure 1 were stable clonal lines or transient pools or stable pools.

Figure 1F- please be clear that the immunoblot panels are of the 786-M1A cells from the top of Figure 1F.

The layout of Figure S2 is confusing and out of order, particularly the placement of S2E

Figure S2B- for the non-786M1A cells that all have multiple bands in the KLF6 western blot, which band are the authors claiming is KLF6. In each the smaller products seem to disappear with KO, but not the larger products. This gets back to how the CRISPR pools were made as well.

Figure S2D- does overexpression of KLF6 reduce growth? Is this why not full rescue? Does the lack of a statistical indicator mean that the KLF6 restored cells are not different from CTRL cells? Was this transiently re-expressed , using stable clones or pool expression?

For Figure 2 please be clear in the text that these experiments were done with the CRISPR/Cas9 knockouts and not the CRISPRi cells.

Please describe CRISPRi approach more in the text. What is being targeted and where to suppress the enhancers.

Figure 4F seems more appropriate to be included in Figure 5 since it is presumably what all the pathway search data is based on.

Reviewer #3 (Remarks to the Author):

KLF6 and lipid metabolism have been the focus of several recently published articles studying ccRCC. This group approached the topic by showing knockout or knockdown of KLF6 obliterated tumor growth in xenograft models, and transcriptomic analysis to develop a signaling pathway around PDGFB and lipid metabolism. Conceptually, the manuscript is easy to follow and the authors clearly describe their rationale for each step. However, the themes feel disconnected from each other and the authors left significant gaps regarding the underlying mechanisms. Specifically, lipid metabolism is offered as a potential mechanism as to how KLF6 supports tumor growth, yet, this is left unexplored. One recently published paper demonstrated opposite results than those presented in this manuscript where KLF6 suppresses tumor growth and invasion, which the

authors did offer discussion on, but missed an opportunity to offer action on. I would suggest that the authors revisit several of their experiments and strengthen their mechanism experiments, as well as offer evidence that can further support their model. Additionally, a group has recently shown that HIF drives lipid deposition by suppressing CPT1A and, here, the authors do suggest that KLF6 is an intermediate between HIF and its targets, so it is my opinion that the authors would need to address whether KLF6 regulates CPT1A. Overall, the results are fairly preliminary and lack mechanistic rigor in several key areas, resulting in incremental novelty over prior studies in the absence of substantial additional work.

Point-by-point response to referee comments

Reviewer #1 (Remarks to the Author)

The manuscript by the Vanharanta lab identifies *KLF6* expression as a *HIF2A* target highly upregulated in renal cell carcinoma cells. They go on and show that *KLF6* expression promotes cancer cell proliferation and they propose that *KLF6* acts by upregulating the mTOR/SREBP pathway through the transcriptional regulation of *PDGFB*. The data are clear and the techniques are sound. I just have few comments:

1- The effect on tumor volume after editing of *KLF6* is minor (Fig. 2A and 2B). Please argue. Are the cells in vivo less dependent on *PDGFB* production? What's the status of the mTOR pathway? Are these tumors resistant to rapamycin?

Reply: There are multiple possible explanations, technical and biological, for the observed effects of *KLF6* inhibition on tumour growth in vivo. In subcutaneous tumour assays with 500k cells injected, we observe a 2-2.7-fold reduction in tumour volume at the end point (**Figure 2a-b**), whereas in an experimental metastasis assay we detect a >10 reduction in overall tumour load in the lungs (**Figure 2c**). This indicates that *KLF6*-dependent pro-tumorigenic functions may be more important for cancer cells under more stringent growth conditions, possibly with lower concentrations of survival factors from neighbouring cells. Another difference between these experiments was that the subcutaneous assays were performed with cells in which *KLF6* was mutated using CRISPR/Cas9, whereas the lung colonization assays was performed with cells in which *KLF6* expression was inhibited using CRISPR interference (CRISPRi). It is possible that the CRISPR/Cas9 approach allows escaper clones to grow out more easily when compared to CRISPRi cells, especially since the subcutaneous tumour assays require more cells, and consequently the cells were propagated longer in tissue culture before the in vivo assays started. To directly test the possibility that at least some of the cells in the subcutaneous tumours may have wildtype *KLF6* activity, we have now extracted genomic DNA from two tumours in which *KLF6* was targeted by CRISPR/Cas9 and one control tumour, and used high-throughput sequencing to evaluate the presence of wildtype *KLF6*. As expected, in the control tumour 99.6% of the reads were wildtype, with the 0.4% of mutant reads most likely resulting from sequencing errors (**Supplementary Figure 3d**). However, when compared to the *KLF6*-targeted cells sampled shortly after viral transduction, in which <3% of the *KLF6* alleles were wildtype (**Supplementary Figure 2a**), the two *KLF6*-targeted tumours had significantly higher fractions of wildtype *KLF6* alleles: 18% and 66%, respectively (**Supplementary Figure 3d**). This indicates that at least a fraction of the observed tumours formed by *KLF6*-targeted cells contained escaper clones. Furthermore, we have now tested the effects of everolimus, a rapamycin analogue, on the growth of M1A xenografts. In line with our model, mTOR inhibition resulted in smaller tumours (**Supplementary Figure 8c**). In conclusion, our in vivo data is in good agreement with the possibility that *KLF6* is an important mediator of ccRCC growth, although it is likely that the effects of *KLF6* inhibition on tumour growth in the subcutaneous may have been underestimated due to escaper clones.

2- The samples in Fig. 6F should be loaded on the same gel and quantified. What is the effect of *PDGFB* rescue on SREBP cleavage and activity?

Reply: (i) The samples in **Figure 6f** were originally loaded on the same gel, and we agree that this should be made clear in the figure. The presentation has now been clarified and the data has been quantified in the revised data panels in **Figure 6f-g**. (ii) We agree that clarifying the degree to which reduced expression of *PDGFB* explains the phenotypic effects of *KLF6* inhibition is important for our study, and we have performed a number of new experiments to address this general point. First, we have found that while mTOR inhibition reduces SREBF1 and SREBF2 activity (**Supplementary Figure 8a**), consequently leading to reduced cholesterol levels (**Supplementary Figure 8b**), mTOR inhibition does not result in a reduction in SREBF2 mRNA expression (**Supplementary Figure 8d**). In contrast, while *KLF6* inhibition reduces mTOR activity (**Figure 6a** and **Supplementary Figure 10a-b**), and this effect can be rescued by *PDGFB* (**Figure 6f-g**), *KLF6* inhibition also results in a reduction in SREBF2 mRNA expression (**Supplementary Figure 6b**). This suggests that in addition to regulating SREBF1 and SREBF2 activity, *KLF6* directly supports at least SREBF2 mRNA expression, but possibly also SREBF1 and/or downstream target gene expression, in an mTOR independent manner. Our new CHIP-seq data supports this possibility with strong *KLF6* peaks observed in the SREBF1 and

SREBF2 upstream regulatory regions, but also upstream of some of the SREBF1 and SREBF2 target genes such as SCD and LSS (**Supplementary Figure 9b-f**). Thus, based on these new data, we don't expect that PDGFB alone can rescue the lipid metabolic or growth phenotypes caused by KLF6 depletion. Collectively, our results suggest that KLF6 regulates lipid metabolism at two levels. First, KLF6 regulates the expression of lipid metabolism genes. Second, KLF6 enhances the activity of SREBF1 and SREBF2 by promoting mTOR activity through PDGFB. These new data, and the consequently revised model in **Figure 7**, are discussed on **pages 13-14** and **16-17** of the revised manuscript.

Reviewer #2 (Remarks to the Author):

The manuscript from Syafruddin et. al. describes a transcription factor controlled signaling axis whereby KLF6 controls lipid homeostasis and tumor growth in clear cell Renal Cell Carcinoma through regulation of PDGFB and mTOR. The authors work helps to address how mTOR signaling is upregulated in many ccRCC tumors even without genetic or mutational alterations to mTOR signaling axis. Understanding this process could help lead to better targeted therapies for ccRCC patients. To address this and identify potential vulnerabilities of ccRCC cells, the authors profiled ccRCC cell lines and identified super enhancers that might regulate key pathways. The authors settled on analysis of KLF6 as the super enhancer regulating its expression was highly acetylated in ccRCC cells and KLF6 levels are upregulated in many ccRCC tumors. The authors characterize KLF6 function through KO and CRISPi to determine that it is important in regulating cell and tumor growth in vitro and in vivo. Through profiling KLF6 depleted cells, the authors identify lipid metabolism and mTOR signaling as key regulated networks and determine that KLF6 up regulates PDGFB which in turn increases mTOR signaling. The authors have the foundations of an important model, but additional work is required to complete the loop.

[1] *The authors should complete the loop and demonstrate that PDGFB is a direct KLF6 target by ChIP. This is a key component to the author's model and it is important to know if PDGFB is a direct target or not. Further the authors should demonstrate that PDGFB is required in cells with high levels of KLF6 for maintaining mTOR signaling.*

Reply: (i) We agree that testing whether KLF6 directly regulates PDGFB expression is an important addition to our study. We have now performed KLF6 ChIP-seq analysis by expressing a flag-tagged KLF6 in KLF6 depleted cells (in the absence of suitable KLF6 antibodies for direct KLF6 ChIP). These data suggest that KLF6 binds directly to the proximal regulatory regions of PDGFB (**Supplementary Figure 11c** and **page 15**), a result confirmed by ChIP-qPCR analysis (**Supplementary Figure 11d** and **page 15**). These data suggest that KLF6 may regulate PDGFB expression directly. (ii) We agree that demonstrating the effects of PDGFB on mTOR activity in ccRCC cells is important. Our study contains two sets of data addressing this point. First, we show that recombinant PDGFB rescues mTOR activity in KLF6 depleted cells (**Figure 6f-g** and **page 16**). Second, we show that inhibition of PDGFB expression reduces mTOR activity (**Figure 6e**, **Supplementary Figure 11f** and **page 16**).

[2] *It is not clear how the lipid metabolism defects are connected to the PDGFB signaling loop. Does PDGFB supplementation restore the cholesterol defects of iKLF6 cells? Is this process mTOR regulated, does inhibiting mTOR block this? Are the cholesterol synthesis and lipid metabolism genes direct KLF6 targets or SREBF1/2 targets and are SREBF1/2 direct KLF6 targets. This information would help complete the signaling loop the authors are describing.*

Reply: We agree that clarifying the connections between KLF6, lipid metabolism and the PDGFB-mTOR axis is important for our study, and we have performed a number of new experiments to address these points. First, we have found that while mTOR inhibition reduces SREBF1 and SREBF2 activity (**Supplementary Figure 8a**), consequently leading to reduced cholesterol levels (**Supplementary Figure 8b**), mTOR inhibition does not result in a reduction in SREBF2 mRNA expression (**Supplementary Figure 8d**). In contrast, while KLF6 inhibition reduces mTOR activity (**Figure 6a** and **Supplementary Figure 10a-b**), and this effect can be rescued by PDGFB (**Figure 6f-g**), KLF6 inhibition also results in a reduction in SREBF2 mRNA expression (**Supplementary Figure 6b**). This suggests that in addition to regulating SREBF1 and SREBF2 activity, KLF6 directly supports at least SREBF2 mRNA expression, but possibly also SREBF1 and/or downstream target gene

expression, in an mTOR independent manner. Our new ChIP-seq data supports this possibility with strong KLF6 peaks observed in the SREBF1 and SREBF2 upstream regulatory regions, but also upstream of some of the SREBF1 and SREBF2 target genes such as SCD and LSS (**Supplementary Figure 9b-f**). Thus, based on these new data, we don't expect that PDGFB alone can rescue the lipid metabolic or growth phenotypes caused by KLF6 depletion. Furthermore, to address whether the predicted SREBF1 and SREBF2 target genes are likely to be directly regulated by SREBF1 and SREBF2, we have analysed publically available SREBF1 and SREBF2 ChIP-seq data sets. As indicated in the pathway map shown in **Supplementary Fig. 6a**, many of the predicted targets contain SREBF1 and SREBF2 binding peaks within their proximal regulatory regions. Collectively, our results suggest that KLF6 regulates lipid metabolism at two levels. First, KLF6 regulates the expression of lipid metabolism genes. Second, KLF6 enhances the activity of SREBF1 and SREBF2 by promoting mTOR activity through PDGFB. These new data, and the consequently revised model in **Figure 7**, are discussed on **pages 13-14** and **16-17** of the revised manuscript.

[3] It is unclear how the super enhancer regulating KLF6 expression is functioning. Introduction of VHL downregulated KLF6 more strongly than any single super enhancer CRISPRi interference, but results in only modest change to a single upstream KLF6 super enhancer element that was not tested by CRISPRi. The authors should test the VHL/HIF2A regulated enhancer by CRISPRi and see if this results in more dramatic KLF6 suppression. While the authors argue that the super enhancer is a robust regulator that withstands interference, it may be that it is not the key or only regulator of KLF6 expression. If the authors combine more than 2 CRISPRi (maybe even all 5) without complete genetic deletion of the SE, does result in strong KLF6 suppression? Were other genes effected strongly by the deletion of the super enhancer (that aren't KLF6 targets)? More data is needed to ensure that super enhancer regulates KLF6 is needed.

Reply: We agree that additional data on the putative *KLF6* super enhancer would improve our study. As suggested, we have now targeted the five previously studied constituent enhancers simultaneously. This led to a further reduction in *KLF6* expression when compared to single or double enhancer targeting (**Figure 3b**). The effect was almost as strong as that seen with the 113kb genetic deletion (**Figure 3c-d**). We have also used CRISPRi to target two putative HIF2A binding sites downstream of *KLF6* in the region showing H3K27ac alterations upon VHL reintroduction, leading to ~40% reduction in *KLF6* expression (**Figure 4f**). This effect is of similar magnitude to that seen by VHL reintroduction, indicating that the VHL-HIF2A pathway supports *KLF6* expression through this segment of the super enhancer. Finally, we have performed RNA-seq analysis comparing cells with the 113kb enhancer deletion to cells with no deletion. This revealed that *KLF6* was the most strongly downregulated gene within a 5Mb genomic region flanking the deleted super enhancer segment (**Figure 3e**). While these analyses do not exclude the possibility that the super enhancer has additional target genes, collectively our results strongly suggests that *KLF6* expression is regulated by this large enhancer cluster. However, we agree that in addition to the super enhancer region, there are other genomic loci that are critical for the expression of *KLF6*. This is reflected, for example, in the fact that direct CRISPRi-mediated targeting of the *KLF6* transcriptional start site results in ~95% reduction in *KLF6* expression (**Supplementary Figure 2e**), a significantly stronger effect than that seen by targeting any of the regulatory regions tested. This is in line with one of the conclusion of our work, which suggests that without functional analysis predicting the significance of gene regulatory elements by chromatin profiling alone is challenging (discussion on **pages 17-18**).

[4] The authors mention the clinical trials testing combos of mTOR and promiscuous VEGF/PDGFR inhibitors in ccRCC cancer and that combo is more effective than single agent therapy. Since these results are known, the authors should consider describing their results as a mechanistic reason to explain the effectiveness of the combination therapy rather than as a rationale for trying it.

Reply: We agree with this suggestion and have revised the abstract, introduction and discussion accordingly.

[5] The authors need to better describe (in legends or text, or both) more about both the traditional CIRSPR and CRISPRi cell lines. It is not clear anywhere if the authors selected individual clones or maintained the pools from initial infections.

Reply: We agree. Apart from the large enhancer deletions, we used pools in all the experiments. This has been clarified in the text and figure legends throughout the manuscript.

Minor issues:

[1] *The authors should be careful about the use of the word autocrine since they provide no evidence that the PDGFB secreted acts on the same cell that secreted it.*

Reply: We agree, using the word 'autocrine' is not appropriate in this context. We have removed the word from the manuscript.

[2] *Figure S1A/B- not clear what the y axis is, is this RPKM?*

Reply: The y axis represents RSEM normalized counts from the TCGA consortium. This has been clarified in the figure legend.

[3] *Figure S1C how about a non-ccRCC line with lower expression? How do we know these cells have elevated KLF6 expression when they are all the same?*

Reply: We agree. We have rerun these samples together with a non-ccRCC cell line as a control (**Supplementary Figure 1c**).

[4] *Please be more clear if the CRISPR/Cas9 knockout clones in figure 1 were stable clonal lines or transient pools or stable pools.*

Reply: These were stable pools generated by lentiviral Cas9 and sgRNA transduction. This has been clarified in the text (**page 7**) and figure legend.

[5] *Figure 1F- please be clear that the immunoblot panels are of the 786-M1A cells from the top of Figure 1F.*

Reply: Yes, the immunoblot data is from the cells used in the proliferation data. This has been clarified in the figure and the figure legend.

[6] *The layout of Figure S2 is confusing and out of order, particularly the placement of S2E*

Reply: We agree, the layout in **Supplementary Figure 2** has been corrected.

[7] *Figure S2B- for the non-786M1A cells that all have multiple bands in the KLF6 western blot, which band are the authors claiming is KLF6. In each the smaller products seem to disappear with KO, but not the larger products. This gets back to how the CRISPR pools were made as well.*

Reply: The cells used in **Supplementary Fig. 2b** are lentivirally generated pools of CRISPR/Cas9 cells. The KLF6 antibody used in these experiments has been validated by CRISPRi knock-down and cDNA over-expression (e.g. **Figure 1e-f**). We are therefore confident that the antibody detects KLF6. However, as pointed out by the referee, especially with higher exposure the antibody detects multiple bands of a fairly similar size, most of which are reduced when KLF6 expression is inhibited by CRISPRi (**Figure 1f, Supplementary Figure 10a-b**). In **Supplementary Figure 2b**, the lower bands in RCC-MF and UOK101 seem to be more prominently depleted, but the higher bands are also reduced, suggesting that they both represent KLF6. In agreement, CRISPRi-mediated KLF6 depletion in the UOK101 cells reduces the expression of several bands detected with this antibody (**Supplementary Figure 10b**). The fact that we observe multiple close bands that disappear upon KLF6 inhibition suggests that alternatively processed KLF6 variants may be expressed, but further studies would be needed for a comprehensive characterization of these variants. We agree that the CRISPR/Cas9 knock-out approach may have variable effects on different KLF6 variants. The detection of multiple possible KLF6 variants was thus one of the reasons that prompted us to favour the CRISPRi approach in the later experiments.

[8] *Figure S2D- does overexpression of KLF6 reduce growth? Is this why not full rescue? Does the*

lack of a statistical indicator mean that the KLF6 restored cells are not different from CTRL cells? Was this transiently re-expressed, using stable clones or pool expression?

Reply: We agree that the effect of KLF6 rescue is not complete in the KLF6 CRISPR/Cas9 cells. Both the KLF6 targeting constructs and the rescue constructs were delivered using lentiviruses, followed by antibiotic selection. There are multiple reasons why a rescue experiment like this may not result in a complete rescue phenotype. For example, technical reasons may result in absent transgene expression in a fraction of KLF6 transduced cells even if the antibiotic resistance gene is expressed. Furthermore, even if all cells express KLF6, the level will vary depending on the viral integration site and it may not be sufficient in each cell, or it may be too high in some cells, as suggested by the referee. Assessing the level of transgene expression on a per cell basis, the level at which the phenotype is determined, is challenging as Western blotting and mRNA analysis measure the cell population as a pool. CRISPR/Cas9 caused mutations may also reduce cell fitness even if the effects of the target gene depletion may be correctly rescued. The fact that the rescue experiment on CRISPRi cells (**Figure 1f**) showed a better rescue effect when compared to the CRISPR/Cas9 cells suggests that this may be another factor contributing to the phenotype seen in **Supplementary Fig. 2d**. Additionally, while we did not see evidence for strong expression of the known alternative transcripts from the *KLF6* locus, endogenous *KLF6* may produce multiple functionally important variants, a possibility that is not accounted for by cDNA expression-based rescue experiments. Finally, it is possible that for optimal function, KLF6 expression needs to be controlled, e.g. depending on the cell cycle, in a way that is not reflected by the cDNA constructs. In sum, multiple technical and biological factors, many of which are very difficult to control for, may complicate genetic rescue experiments in general. However, we have used two different methods of KLF6 inhibition, with two targeting constructs each, combined with rescue experiments that show a clear rescue effect, especially in the early time point of the CRISPRi experiment (**Figure 1f**). We are therefore confident that the vast majority of phenotypic effects we observe upon KLF6 inhibition are due to KLF6 inhibition. We have added p-values for all the experimental comparisons in the **Supplementary Fig. 2d**.

[9] For Figure 2 please be clear in the text that these experiments were done with the CRISPR/Cas9 knockouts and not the CRispri cells.

Reply: The experiments in **Figure 2a-b** have been conducted with lentivirally generated pools of CRISPR/Cas9 cells. The experiment in **Fig. 2c-d** was conducted with lentivirally generated pools of CRISPRi cells. This has been clarified in the figure legend.

[10] Please describe CRISPRi approach more in the text. What is being targeted and where to suppress the enhancers.

Reply: We target enhancers by designing guide RNAs that target p300 peaks adjacent to the H3K27ac regions, as transcription factors are in general considered to bind in these 'valleys' within H3K27ac regions. We have clarified this approach in the text on **page 8**.

[11] Figure 4F seems more appropriate to be included in Figure 5 since it is presumably what all the pathway search data is based on.

Reply: We agree. We have reorganised the data panels accordingly.

Reviewer #3 (Remarks to the Author):

KLF6 and lipid metabolism have been the focus of several recently published articles studying ccRCC. This group approached the topic by showing knockout or knockdown of KLF6 obliterated tumor growth in xenograft models, and transcriptomic analysis to develop a signaling pathway around PDGFB and lipid metabolism. Conceptually, the manuscript is easy to follow and the authors clearly describe their rationale for each step.

[1] However, the themes feel disconnected from each other and the authors left significant gaps regarding the underlying mechanisms. Specifically, lipid metabolism is offered as a potential mechanism as to how KLF6 supports tumor growth, yet, this is left unexplored.

Reply: We agree that strengthening the mechanistic links between KLF6, lipid metabolism and ccRCC growth would improve our study. To this end, we have now added several pieces of new data showing (i) that mTOR inhibition reduces cellular cholesterol levels in ccRCC cells, phenocopying the effects of KLF6 inhibition (**Supplementary Figure 8b** and **page 14**), (ii) that genetic SREBF1/2 inhibition reduces cellular cholesterol levels (**Figure 5e**, **Supplementary Figure 7a** and **page 13**) and impairs ccRCC growth (**Figure 5f** and **page 13**), also phenocopying KLF6 inhibition, and (iii) that pharmacological inhibition of SREBF1/2 using fatostatin or cholesterol biosynthesis by simvastatin inhibit ccRCC growth (**Figure 5g**, **Supplementary Figure 7b** and **page 13**), further strengthening the link between lipid homeostasis and ccRCC growth. Thus, while these and other experiments in our study do not exclude the possibility that KLF6 supports ccRCC growth through additional mechanisms, our data strongly suggests that inhibition of KLF6-dependent lipid metabolism is a major contributor to the effects observed upon KLF6 inhibition. We have revised the model in **Figure 7** to highlight these new results.

[2] I would suggest that the authors revisit several of their experiments and strengthen their mechanism experiments, as well as offer evidence that can further support their model.

Reply: We agree that further experimental evidence would strengthen our model. Therefore, in addition to the new results described above (reply to Reviewer #3 point [1]), which link the growth defects of KLF6 inhibited cells to reduced lipid metabolism, the revised manuscript contains several new data sets that clarify important points related to the role of KLF6 in ccRCC. Most importantly, the new results demonstrate that KLF6 regulates lipid homeostasis at two levels. First, KLF6 regulates the expression of lipid metabolism genes. Second, via regulating PDGFB and consequently supporting mTOR signalling, KLF6 promotes the activity of SREBF1 and SREBF2. Specifically, we find that mTOR inhibition reduces the expression of activated SREBF1 and SREBF2 (**Supplementary Figure 8a** and **page 14**). However, in contrast to KLF6 inhibition, which results in reduced expression of SREBF1, SREBF2 and their target genes (**Supplementary Figure 6a-b** and **pages 12-13**), mTOR inhibition reduces the expression of SREBF1 and the SREBF1/2 target genes, but not the expression of SREBF2 mRNA (**Supplementary Figure 8d** and **page 14**). In agreement with the possible direct role of KLF6 as a regulator of SREBF1 and SREBF2 expression, new ChIP-seq data demonstrate prominent KLF6 peaks at the proximal regulatory regions of these genes (**Supplementary Figure 9b-f** and **pages 14-15**). Additionally, we provide new data on the role of the VHL-HIF2A pathway as a regulator of KLF6 expression through a distal enhancer element that is part of the large super enhancer flanking the *KLF6* locus (**Figure 4f** and **pages 10-11**). New RNA-seq data on the effects of the large 113kb super enhancer deletion demonstrate that *KLF6* is a major target gene for this enhancer cluster (**Figure 3e** and **pages 9-10**), and new data on combinatorial targeting of several constituent enhancers of the super enhancer locus further strengthen their role as regulators of *KLF6* (**Figure 3b** and **page 9**). Furthermore, the revised manuscript has been edited throughout to provide additional technical and conceptual clarification as well as generally improved presentation. In sum, the paper has been significantly improved and it collectively provides strong support for the proposed model of KLF6 as a mediator of ccRCC growth. The newly discovered dual role of KLF6 as a regulator of lipid metabolism, as well as the new data demonstrating the effects of SREBF1 and SREBF2 on ccRCC growth are highlighted in the revised model in **Figure 7**.

[3] One recently published paper demonstrated opposite results than those presented in this manuscript where KLF6 suppresses tumor growth and invasion, which the authors did offer discussion on, but missed an opportunity to offer action on.

Reply: Gao et al. have indeed recently published a paper suggesting that KLF6 acts as a suppressor of metastasis in ccRCC (*Cancer Res* 77, 330-342). As we discuss on **pages 18-19**, we do not understand the reasons for the differences between our results and those by Gao et al. We do note, however, that in contrast to our work, which uses two independent sgRNA constructs in both CRISPRi and CRISPR/Cas9 mutagenesis-based gene inactivation settings, combined with genetic rescue experiments, the loss-of-function analyses of Gao et al. rely solely on a single RNAi sequence, making it difficult to exclude off-target effects. Mechanistically, Gao et al. report that *KLF6* represses the expression of *E2F1*. In contrast, using RNA-seq and validation by qRT-PCR, we find that KLF6 supports *E2F1* expression (**Supplementary Figure 12a**). We therefore conclude that the discrepancies between our study and that by Gao et al. are likely due to technical issues, and we don't see obvious further opportunities for additional analysis at our end.

[4] *Additionally, a group has recently shown that HIF drives lipid deposition by suppressing CPT1A and, here, the authors do suggest that KLF6 is an intermediate between HIF and its targets, so it is my opinion that the authors would need to address whether KLF6 regulates CPT1A.*

Reply: We agree that linking the present results with existing literature is important. Our RNA-seq data, now confirmed by qRT-PCR (**Supplementary Figure 12b**), show that *CPT1A* expression is downregulated upon KLF6 depletion. This suggests that in addition to regulating the SREBF1 and SREBF2 pathways, KLF6-dependent effects on ccRCC lipid metabolism may also involve CPT1A. However, as VHL reintroduction increases CPT1A expression but decreases KLF6 expression, and KLF6 supports CPT1A expression, it seems unlikely that the pro-tumorigenic effects of KLF6 are directly caused by modulation of CPT1A activity through the mechanism described in the study by Du et al. (Nat Commun. 2017 Nov 24;8(1):1769). Further work is thus needed for a comprehensive understanding of how different mediators of lipid metabolic phenotypes contribute to ccRCC. We have added a paragraph on **page 19** to discuss this point.

[5] *Overall, the results are fairly preliminary and lack mechanistic rigor in several key areas, resulting in incremental novelty over prior studies in the absence of substantial additional work.*

Reply: As discussed above (Reviewer #3 points [1]-[2]), the revised manuscript contains extensive data that give strong support to the proposed model which describes the mechanism through which KLF6 supports ccRCC growth by promoting lipid homeostasis at two levels: (i) by regulating the expression of lipid metabolism genes, and (ii) by regulating the activity of SREBF1 and SREBF2 through the PDGFB-mTOR axis. This mode of KLF6 action has not been previously described. The mTOR and PDGFB pathways are clinically approved targets in ccRCC, but the mechanisms through which they get activated in ccRCC have remained elusive. Our results thus provide new molecular insight into the mechanisms that activate these pathways in ccRCC, giving a possible explanation for the favourable outcome seen in ccRCC patients treated with combinatorial mTOR/RTK inhibition (Motzer, R. J. et al. *Lancet Oncol* 16, 1473-1482). Furthermore, the general concept emerging from our work, that super enhancer-driven tissue-specific transcriptional networks are needed for the activity of core metabolic pathways gives clues to the mechanisms that may support the stability of cell identity-defining transcriptional networks in cancer. Thus, our results are novel at the specific mechanistic level, but also at the more general conceptual level, and we are not aware of similar results having been published elsewhere.

Reviewers' comments:

Reviewer #1 (Remarks to the Author):

The authors addressed all the issues previously raised.

Mario Pende

Reviewer #2 (Remarks to the Author):

The authors have provided substantial new data that address most of my concerns. The authors have now included experiments which provide connections between lipid metabolism, HIF2A and mTOR and how changes in KLF6 super enhancer activity affect and interrelate these processes. In addition the experimental details and statistical methods are much more thorough in the revised manuscript.

Minor note- Figure 3E- X axis- fix the $-\log$ scale it says -15 where it should presumably be -5.

Reviewer #3 (Remarks to the Author):

The authors of this manuscript have made several additions to strengthen their underlying assertions that KLF6 is regulated via a HIF2 expression driven superenhancer to modulate tumor metabolic properties, specifically demonstrated by alterations of mTOR signaling. Significant improvements in labeling figures and figure placement have been made, and concessions on the sole focus on a single cell line have been presented. Although the genomic work is intriguing, this reviewer remains concerned that the overall link to a VHL-mutation-driven biology is lacking. The experiments focused on HIF2 linkage remain limited to the 786-0 M1A (a highly manipulated line to make it have metastatic potential, with demonstrated epigenetic reprogramming in the manuscript in which these cells were derived), and do not demonstrate functional changes to signaling or tumor cell assays that assure the effects are not independent (effects of KLF6 knockdown/knockout in the setting of HIF2 suppression or VHL add back--in a cell line other than 786-0 M1A). The concession of lining up a panel of VHL mutant tumor lines against a lung cancer line, without showing VHL add back is not sufficient.

If this paper remains highly focused on 786-0 M1A, a further discussion of the unique properties of the cell line is needed.

Point-by-point response to referee comments

Reviewer #1 (Remarks to the Author)

“The authors addressed all the issues previously raised.”

Reviewer #2 (Remarks to the Author)

“The authors have provided substantial new data that address most of my concerns. The authors have now included experiments which provide connections between lipid metabolism, HIF2A and mTOR and how changes in KLF6 super enhancer activity affect and interrelate these processes. In addition the experimental details and statistical methods are much more thorough in the revised manuscript.

Minor note- Figure 3E- X axis- fix the –log scale it says -15 where it should presumably be -5.”

Reply: Thank you for pointing this out, we have corrected the error.

Reviewer #3 (Remarks to the Author)

“The authors of this manuscript have made several additions to strengthen their underlying assertions that KLF6 is regulated via a HIF2 expression driven superenhancer to modulate tumor metabolic properties, specifically demonstrated by alterations of mTOR signaling. Significant improvements in labeling figures and figure placement have been made, and concessions on the sole focus on a single cell line have been presented. Although the genomic work is intriguing, this reviewer remains concerned that the overall link to a VHL-mutation-driven biology is lacking. The experiments focused on HIF2 linkage remain limited to the 786-0 M1A (a highly manipulated line to make it have metastatic potential, with demonstrated epigenetic reprogramming in the manuscript in which these cells were derived), and do not demonstrate functional changes to signaling or tumor cell assays that assure the effects are not independent (effects of KLF6 knockdown/knockout in the setting of HIF2 suppression or VHL add back--in a cell line other than 786-0 M1A). The concession of lining up a panel of VHL mutant tumor lines against a lung cancer line, without showing VHL add back is not sufficient.”

“If this paper remains highly focused on 786-0 M1A, a further discussion of the unique properties of the cell line is needed.”

Reply: We agree that it is important to validate experimental findings in multiple cell lines. For this reason, our initial characterisation of the KLF6 phenotype was performed in four different ccRCC cell lines (**Figure 1d-f, Supplementary Fig. 2b-d**), with *in vivo* validation using two different cell lines (**Figure 2a-d, Supplementary Fig. 3a-d**). Additionally, the link between ccRCC and high expression of KLF6 mRNA is supported by the analysis of ~7,000 cancer samples from the TCGA data set (**Supplementary Fig. 1a**). Given the specificity of *VHL* mutations to ccRCC, and the fact that up to 90% of ccRCCs carry inactivating alterations in the *VHL* pathway [1], the TCGA data set gives very strong genetic evidence for the link between *VHL* inactivation and high *KLF6* expression. The link between high HIF2A activity in ccRCC is further corroborated by the data showing a strong correlation between *HIF2A* and *KLF6* mRNA levels in ~500 ccRCC specimens in the TCGA data set (**Figure 4a**). Furthermore, we show in two cell lines that *VHL* restoration results in *KLF6* mRNA downregulation (**Figure 4b**) and reduced activity of a constituent enhancer of the large *KLF6* super enhancer (**Figure 4d**). Moreover, this constituent enhancer binds HIF2A in both the 786-M1A cells (**Figure 4e**) and OS-LM1 cells (**new analysis shown in Supplementary Fig. 5a**). We also show a Western blot of the *VHL*-restored cells in **Figure 4c**. Finally, we provide functional data from three different cell lines showing the effects of *KLF6* inhibition on PDGFB mRNA expression and mTOR activity (**Figure 6a-b, Figure 6e-g, and Supplementary Fig. 10a-d**), with supporting evidence from the TCGA ccRCC data set (**Figure 6c-d**). Thus, contrary to what the referee implies, our conclusions are based on data from several different cell lines with validation in large clinical data sets.

As discussed above, our study is not focused on a single cell line, but rather on the biology of human ccRCC. However, we do agree that experimental systems and their clinical relevance should be accurately reported. As described in our published work [2], the 786-M1A cell line was derived via

a single round of *in vivo* selection for 786-O subclones that are capable of robust lung colonisation in mice; it is not a “highly manipulated line” as suggested by the referee. Importantly, these cells have several characteristics that support their use as a clinically relevant model of advanced ccRCC. Specifically, the 786-M1A cells (i) harbour mutations in genes that are frequently inactivated in human ccRCC [3], (ii) they form tumours in mice that histologically resemble aggressive human ccRCC [2], (iii) they display high metastatic potential in several different *in vivo* assays to organs affected by ccRCC in human patients [2, 4], (iv) they express genes that correlate with poor patient outcome in clinical data sets [2, 4], (v) their gene regulatory landscape correlates with transcription in a manner similar to that seen in clinical data sets [4], and (vi) they respond to treatment with HIF2A [5] and mTOR inhibitors (**Supplementary Fig. 8c**), mimicking the clinical behaviour of human ccRCC. As requested, we have added a note on the model systems on **page 6** of the revised manuscript.

1. Turajlic, S., et al., *Deterministic Evolutionary Trajectories Influence Primary Tumor Growth: TRACERx Renal*. Cell, 2018. **173**(3): p. 595-610 e11.
2. Vanharanta, S., et al., *Epigenetic expansion of VHL-HIF signal output drives multiorgan metastasis in renal cancer*. Nat Med, 2013. **19**(1): p. 50-6.
3. Jacob, L.S., et al., *Metastatic Competence Can Emerge with Selection of Preexisting Oncogenic Alleles without a Need of New Mutations*. Cancer Res, 2015. **75**(18): p. 3713-9.
4. Rodrigues, P., et al., *NF-kappaB-Dependent Lymphoid Enhancer Co-option Promotes Renal Carcinoma Metastasis*. Cancer Discov, 2018. **8**(7): p. 850-865.
5. Cho, H., et al., *On-target efficacy of a HIF-2alpha antagonist in preclinical kidney cancer models*. Nature, 2016. **539**(7627): p. 107-111.

Point-by-point response to referee comments

Reviewer #1 (Remarks to the Author)

“The authors addressed all the issues previously raised.”

Reviewer #2 (Remarks to the Author)

“The authors have provided substantial new data that address most of my concerns. The authors have now included experiments which provide connections between lipid metabolism, HIF2A and mTOR and how changes in KLF6 super enhancer activity affect and interrelate these processes. In addition the experimental details and statistical methods are much more thorough in the revised manuscript.

Minor note- Figure 3E- X axis- fix the –log scale it says -15 where it should presumably be -5.”

Reply: Thank you for pointing this out, we have corrected the error.

Reviewer #3 (Remarks to the Author)

“The authors of this manuscript have made several additions to strengthen their underlying assertions that KLF6 is regulated via a HIF2 expression driven superenhancer to modulate tumor metabolic properties, specifically demonstrated by alterations of mTOR signaling. Significant improvements in labeling figures and figure placement have been made, and...”

Point 1. *“... concessions on the sole focus on a single cell line have been presented.”*

Reply: We agree that it is important to validate experimental findings in multiple cell lines. For this reason, our initial characterisation of the KLF6 phenotype was performed in four different ccRCC cell lines (**Figure 1d-f, Supplementary Fig. 2b-d**), with *in vivo* validation using two different cell lines (**Figure 2a-d, Supplementary Fig. 3a-d**). Furthermore, we show in two cell lines that *VHL* restoration results in KLF6 mRNA (**Figure 4b**) and protein downregulation (**Figure 4c**, which contains **new data for OS-LM1 cells**), as well as reduced activity of a constituent enhancer of the large KLF6 super enhancer (**Figure 4d**). Moreover, this constituent enhancer binds HIF2A in both the 786-M1A cells (**Figure 4e**) and OS-LM1 cells (**new analysis in Supplementary Fig. 5a**). We also show in two cell lines that *VHL* restoration leads to reduced HIF2A binding at the KLF6 super enhancer locus (**new data in Supplementary Fig. 5b**). Finally, we provide functional data on three different cell lines showing the effects of KLF6 inhibition on PDGFB mRNA expression and mTOR activity (**Figure 6a-c, Figure 6g-h, and Supplementary Fig. 11a-b**), with supporting evidence from the TCGA ccRCC data set (**Figure 6d-e**). Thus, our work is not focused on a single cell line. Rather, our conclusions are based on data from several different cell lines with validation in large clinical data sets. To clarify this point, we have rearranged the data in **Figure 6** and **Supplementary Fig. 11** as well as amended the text on **pages 10, 11** and **15**. The experimental models are also described in more detail on **page 7** of the revised manuscript.

Point 2. *“Although the genomic work is intriguing, this reviewer remains concerned that the overall link to a VHL-mutation-driven biology is lacking. The experiments focused on HIF2 linkage remain limited to the 786-0 M1A...”*

The link between ccRCC and high expression of KLF6 mRNA is supported by the analysis of ~7,000 cancer samples from the TCGA data set (**Supplementary Fig. 1a**). Given the specificity of *VHL* mutations to ccRCC, and the fact that up to 90% of ccRCCs carry inactivating alterations in the *VHL* pathway [1], the TCGA data set gives strong genetic evidence for the link between *VHL* inactivation and high *KLF6* expression. The link between high HIF2A activity in ccRCC is further corroborated by the data showing a strong correlation between *HIF2A* and *KLF6* mRNA levels in ~500 ccRCC specimens in the TCGA data set (**Figure 4a**). Furthermore, we show in two cell lines that *VHL* restoration results in KLF6 mRNA (**Figure 4b**) and protein downregulation (**Figure 4c**, which contains **new data for OS-LM1 cells**), as well as reduced activity of a constituent enhancer of the large KLF6

super enhancer (**Figure 4d**). Moreover, this constituent enhancer binds HIF2A in both 786-M1A cells (**Figure 4e**) and OS-LM1 cells (**new analysis in Supplementary Fig. 5a**). Finally, new ChIP-qPCR analysis demonstrates reduced HIF2A binding at the KLF6 super enhancer locus upon VHL restoration in two cell lines (**new data in Supplementary Fig. 5b**). Collectively, these data give strong support for our model whereby *KLF6* expression is partially regulated by the tumour-initiating VHL-HIF2A pathway in ccRCC cells.

Point 3. *“(a highly manipulated line to make it have metastatic potential, with demonstrated epigenetic reprogramming in the manuscript in which these cells were derived),”*

As described in our published work [2], the 786-M1A cell line was derived via a single round of *in vivo* selection, without further manipulation, for 786-O subclones that are capable of robust lung colonisation in mice. Importantly, these cells have several characteristics that support their use as a relevant model of advanced ccRCC. Specifically, the 786-M1A cells (i) harbour mutations in genes that are inactivated in human ccRCC [3], (ii) they form tumours in mice that histologically resemble aggressive human ccRCC [2], (iii) they display high metastatic potential in several different *in vivo* assays to organs affected by ccRCC in human patients [2, 4], (iv) they express genes that correlate with poor patient outcome in clinical data sets [2, 4], (v) their gene regulatory landscape correlates with transcription in a manner similar to that seen in clinical data sets [4], and (vi) they respond to treatment with HIF2A [5] and mTOR inhibitors (**Supplementary Fig. 9c**), mimicking the clinical behaviour of human ccRCC. Thus, while the metastatic phenotype in these cells is functionally driven by epigenetic reprogramming, the epigenetic changes are not caused by “manipulation”. Rather, they reflect phenotypic evolution of ccRCC towards a more malignant state. To clarify this point, we have included additional information on the model systems on **page 7** of the revised manuscript.

Point 4. *“... and do not demonstrate functional changes to signaling or tumor cell assays that assure the effects are not independent (effects of KLF6 knockdown/knockout in the setting of HIF2 suppression or VHL add back--in a cell line other than 786-0 M1A).”*

Gene set enrichment analysis revealed that genes regulated by KLF6 are enriched in known hypoxia-signalling target genes (**Supplementary Fig. 6a**), and we show that for some genes, such as *BHLHE40*, the negative effect on transcription caused by VHL reintroduction can be rescued by simultaneous expression of KLF6 (**Supplementary Fig. 6b**), a result now reproduced in a second cell line (**new data in Supplementary Fig. 6c**). Additionally, KLF6 knockdown reduces *BHLHE40* expression (**Supplementary Fig. 6d**, containing **new data for OS-LM1 cells**). These data demonstrate interaction between the HIF2A and KLF6 pathways at the level of downstream targets. We agree, however, that while these results are compatible with the possibility that the effect of the VHL-HIF2A pathway on *BHLHE40* expression is dependent on KLF6, the data do not exclude the possibility that HIF2A and KLF6 regulate *BHLHE40* independently. Similarly, as VHL restoration does not completely abrogate KLF6 activity, but leads to ~50% reduction in KLF6 expression, KLF6 knockdown in VHL restored cells would also not conclusively differentiate between KLF6-dependent and KLF6-independent effects of VHL restoration. We agree that this is an important clarification and we have amended the text on **pages 11** and **12** of the revised manuscript accordingly.

Point 5. *“The concession of lining up a panel of VHL mutant tumor lines against a lung cancer line, without showing VHL add back is not sufficient.”*

We agree that the VHL restoration experiment is important. We have performed it in two different cell lines, both of which demonstrate reduced HIF2A and KLF6 protein expression upon VHL reintroduction (**Figure 4c**, containing **new data for OS-LM1 cells**).

Point 6. *“If this paper remains highly focused on 786-0 M1A, a further discussion of the unique properties of the cell line is needed.”*

Our study is not focused on a single cell line, but rather on the biology of human ccRCC, and our conclusions are based on data from several different cell lines with validation in large clinical data sets (please see our response to Point 1 above for details). To further emphasise this point, the revised manuscript contains **new data** on an additional cell line in **Figure 4c** and **Supplementary Fig. 5**. We have also rearranged the data in **Figure 6** and **Supplementary Fig. 11** as well as amended the text on **pages 10, 11** and **15** to highlight the results from different cell lines used in our work. Finally, we

agree that experimental systems and their clinical relevance should be accurately discussed. We have therefore included additional discussion of the cell line models on **page 7** of the revised manuscript.

1. Turajlic, S., et al., *Deterministic Evolutionary Trajectories Influence Primary Tumor Growth: TRACERx Renal*. Cell, 2018. **173**(3): p. 595-610 e11.
2. Vanharanta, S., et al., *Epigenetic expansion of VHL-HIF signal output drives multiorgan metastasis in renal cancer*. Nat Med, 2013. **19**(1): p. 50-6.
3. Jacob, L.S., et al., *Metastatic Competence Can Emerge with Selection of Preexisting Oncogenic Alleles without a Need of New Mutations*. Cancer Res, 2015. **75**(18): p. 3713-9.
4. Rodrigues, P., et al., *NF-kappaB-Dependent Lymphoid Enhancer Co-option Promotes Renal Carcinoma Metastasis*. Cancer Discov, 2018. **8**(7): p. 850-865.
5. Cho, H., et al., *On-target efficacy of a HIF-2alpha antagonist in preclinical kidney cancer models*. Nature, 2016. **539**(7627): p. 107-111.

REVIEWERS' COMMENTS:

Reviewer #3 (Remarks to the Author):

The authors have satisfactorily responded to my concerns.

Response to referee comments.

Reviewer #3 (Remarks to the Author):

The authors have satisfactorily responded to my concerns.

Reply: Thank you for taking the time to review our work.